# CrossGET: Cross-Guided Ensemble of Tokens for Accelerating Vision-Language Transformers

## Abstract

Recent vision-language models have achieved tremendous progress far beyond what we ever expected. However, their computational costs are also dramatically growing with rapid development, especially for the large models. It makes model acceleration exceedingly critical in a scenario of limited resources. Although extensively studied for unimodal models, the acceleration for multimodal models, especially the vision-language Transformers, is relatively under-explored. To pursue more efficient and accessible vision-language Transformers, this paper introduces **Cross-G**uided **E**nsemble of **T**okens (*CrossGET*), a universal acceleration framework for vision-language Transformers. This framework adaptively combines tokens through real-time, cross-modal guidance, thereby achieving substantial acceleration while keeping high performance. *CrossGET* has two key innovations: 1) *Cross-Guided Matching and Ensemble*. *CrossGET* incorporates cross-modal guided token matching and ensemble to exploit cross-modal information effectively, only introducing cross-modal tokens with negligible extra parameters. 2) *Complete-Graph Soft Matching*. In contrast to the existing bipartite soft matching approach, *CrossGET* introduces a complete-graph soft matching policy to achieve more reliable token-matching results while maintaining parallelizability and high efficiency. Extensive experiments are conducted on various vision-language tasks, including image-text retrieval, visual reasoning, image captioning, and visual question answering. Performance on both classic multimodal architectures and emerging multimodal LLMs demonstrate the effectiveness and versatility of the proposed *CrossGET* framework. The code and models will be made public.

## 1 Introduction

The AI community is witnessing the bloom of vision-language models (Kiros et al., 2014; Karpathy et al., 2014; Antol et al., 2015; Vinyals et al., 2015; Yang et al., 2016; Huang et al., 2017; Radford et al., 2021; Wang et al., 2022a; Li et al., 2022; OpenAI, 2023a; Li et al., 2023a), among which transformer-based models such as CLIP (Radford et al., 2021), BLIP (Li et al., 2022), and GPT-4 (OpenAI, 2023b) have become dominant in recent studies. They can tackle a broad range of vision-language tasks, such as Image-Text Retrieval (Jia et al., 2015), Vision Reasoning (Suhr et al., 2018), Image Captioning (Lin et al., 2014), and Visual Question Answer (Antol et al., 2015). Nevertheless, the notable improvement in model performance is at the expense of significantly increased computational cost. High computational requirements are preventing researchers with limited resources and consumers with low-end devices from enjoying the advances of vision-language Transformers.

Numerous model acceleration techniques exist, including token reduction (Chen et al., 2021; Rao et al., 2021; Su et al., 2022; Chavan et al., 2022; Liang et al., 2022b; Yin et al., 2022; Bolya et al., 2023), model pruning (Han et al., 2015; He et al., 2017; Fan et al., 2019; Michel et al., 2019; Zhu et al., 2021; Chavan et al., 2022), quantization (Xiao et al., 2022; Tao et al., 2022; Frantar & Alistarh, 2022; Yuan et al., 2023; Frantar et al., 2023), and knowledge distillation (Hinton et al., 2015; Zhang et al., 2019; Jiao et al., 2019; Wang et al., 2020b; Touvron et al., 2021; Yang et al., 2022). Although widely studied on unimodal models, there still exists a broad space in multimodal scenarios, *e.g.*, accelerating vision-language Transformers. A few works have tried it by model pruning (Shi et al.,

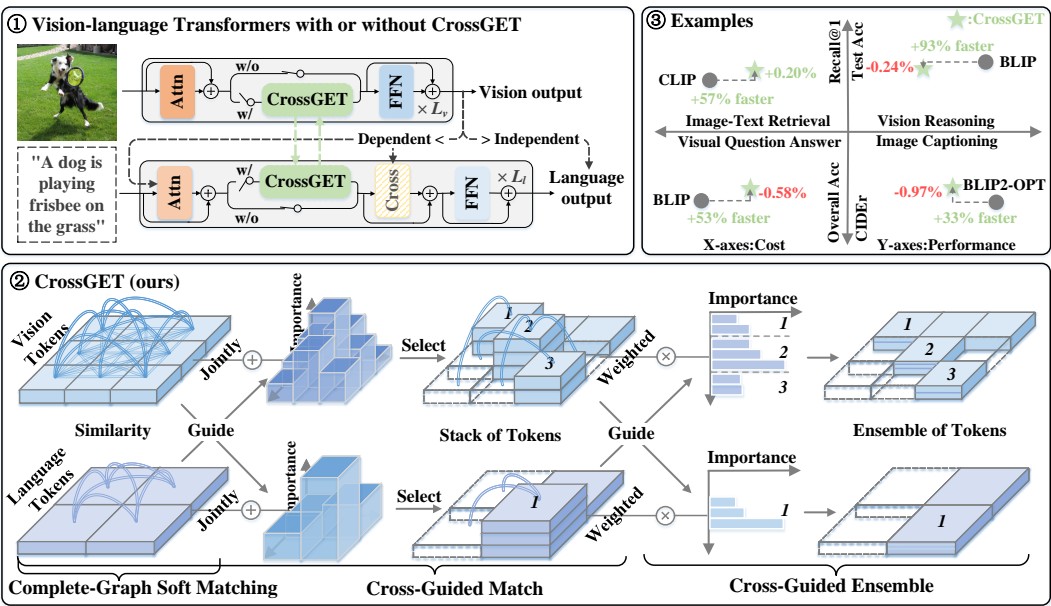

Figure 1: Overview of *CrossGET*. (1) *CrossGET* is a universal multimodal token reduction framework that can be applied to both modality-independent and modality-dependent models. (2) *CrossGET* jointly consider the token similarity given by intra-modal complete-graph soft matching and token importance given by cross-modal guidance to determine which tokens should be combined. The cross-modal importance will be used again to weight tokens within each stack to output their ensembles. (3) Compared to original models, *CrossGET* achieves considerable computation saving and acceleration with negligible performance degradation.

2023) or knowledge distillation (Fang et al., 2021). However, these existing approaches mainly focus on reducing the parameters of models and usually yield less speedup than token reduction.

To bridge this gap, we introduce *CrossGET*, a framework designed to efficiently reduce the token count during inference in vision-language Transformers. *CrossGET* features two primary innovations, *i.e.*, *cross-guided matching and ensemble* and *complete-graph soft matching*.

First, *CrossGET* introduces *cross-guided matching and ensemble* to determine and ensemble redundant tokens. It is applicable for both modality-independent models (*e.g.*, CLIP (Radford et al., 2021)) and modality-dependent models (*e.g.*, BLIP/BLIP2 (Li et al., 2022; 2023b)). *CrossGET* injects cross tokens with negligible parameters into both vision and language branches, letting them learn cross-modal importance and guide the selection of redundant tokens.[1] Besides, the cross-modal importance also serves as the guidance to combine tokens in a weighted ensemble manner.

Second, the existing work adopts a bipartite soft matching approach (Bolya et al., 2023), which splits tokens into two disjoint sets alternately. It is easier to ensure parallelizability but leads to less reliable matching results. To improve matching results, *CrossGET* combines similar tokens in a complete graph whose accurate solution is originally non-parallelizable, and *CrossGET* introduces *complete-graph soft matching* as an approximate solution to ensure parallelizability and high efficiency further.

The contributions of the proposed *CrossGET* framework can be summarized as

- It is one of the early and pioneering works on token ensemble framework for vision-language Transformers, achieving a better performance-efficiency trade-off with negligible extra parameters than previous methods, and for the first time being simultaneously applicable to both modality-independent and modality-dependent vision-language Transformers.

- It proposes *cross-guided matching and ensemble* for effectively exploiting bidirectional cross-modal information, which is more feasible than existing unidirectional approaches,

---

[1]A naive solution is directly calculating the similarity between vision and language tokens. However, in modality-dependent models, the different modality branches are aligned in cross or sequential ways. Therefore, the cross-modal similarity is not available for front modality branches. *CrossGET* lets cross tokens in each modality serve as agents for other modalities, providing cross-modal guidance free from inference.

and *complete-graph soft matching* for more reliable token-matching results than existing bipartite approaches while maintaining parallelizability and high efficiency.

- Its effectiveness and versatility are validated on a broad range of vision-language tasks, datasets, and model architectures. It is also the first time that token ensemble approaches have been applied to the modality-dependent pipeline of BLIP2 (Li et al., 2023b), which is one of the most popular paradigms followed by recent multimodal LLMs, *e.g.*, InstructBLIP (Dai et al., 2023), MiniGPT-4 (Zhu et al., 2023), mPLUG-Owl (Ye et al., 2023), and so on.

## 2 RELATED WORK

**Vision-Language Transoformers**  According to dependency on the computation order of different modalities, the existing vision-language Transformers can be summarized into two categories: 1) modality-independent models (Li et al., 2020; 2021; Radford et al., 2021; Kim et al., 2021; Singh et al., 2022). For example, CLIP (Radford et al., 2021) is a representative model. It is feasible to compute the visual branch first as well as the language branch first. 2) modality-dependent models (Li et al., 2021; Yu et al., 2022; Li et al., 2022; Alayrac et al., 2022), for example BLIP-based models (Li et al., 2022) and BLIP2/LLaVA-based (Li et al., 2023b; Zhu et al., 2023; Dai et al., 2023; Liu et al., 2023; Gao et al., 2023) multimodal LLMs. It is only feasible to compute the vision branch first since the language branch takes vision outputs as part of its inputs. *CrossGET* is a universal framework that can be used for both modality-independent and modality-dependent scenarios.

**Multimodal Transformer Acceleration**  A few works have attempted to accelerate multimodal Transformers. Gan et al. (2022) investigate unstructured pruning and find winning tickets (Frankle & Carbin, 2018) also exist in multimodal Transformers. By structured pruning, UPop (Shi et al., 2023) proposes that small vision-language models can be unifiedly searched in large ones and then progressively pruned. DistillVLM (Fang et al., 2021) and EfficientVLM (Wang et al., 2022b) suggest knowledge distillation to mimic the distribution of large vision-language models. MiniVLM (Wang et al., 2020a) uses lightweight networks to construct its model, and AWQ (Lin et al., 2023) applies weight-only quantization on multimodal Transformers. *CrossGET* achieves acceleration by reducing tokens, which is orthogonal to these existing approaches by shrinking model parameters. TRIPS (Jiang et al., 2022) uses text information to guide the reduction of image patches unidirectionally for pretraining, while CrossGET allows modalities to learn guidance information bidirectionally from each other for fine-tuning. Appendix C.10 provides more comparisons and analyses on TRIPS.

**More Related Works**  More discussions of related works can refer to Appendix B.

## 3 METHODOLOGY

### 3.1 OVERVIEW

Figure 1 illustrates that *CrossGET* accelerates vision-language Transformers by ensembling tokens. It is inserted into the middle of the Self-Attention and FFN of Transformer layers in both the vision and language branches. To achieve the goal of effectively exploiting cross-modal information, *CrossGET* proposes *cross-guided matching and ensemble* (Section 3.2 and 3.3). To achieve more reliable token-matching, *CrossGET* designs a parallelizable *complete-graph soft matching* (Section 3.4).

### 3.2 CROSS-GUIDED MATCHING (CGM)

**Exploiting Cross-Modal Guidance**  For multimodal models, in addition to using the intra-modal similarity as guidance, the token-matching results can also benefit from cross-modal guidance. However, effectively introducing cross-modal guidance is challenging, especially when a dependency exists on the computation order of different modalities.

Intuitively, suppose modality1 needs guidance from modality2 (*e.g.*, to help determine the matching of tokens), then modality2 should conduct inference, output features as cross-modal guidance, and send to modality1. However, if there is a computational dependency, *e.g.*, the output of modality1 is a necessary input for modality2, modality2 cannot conduct inference before modality1 finishes its inference. Therefore, modality1 is unable to exploit cross-modal guidance provided by modality2.

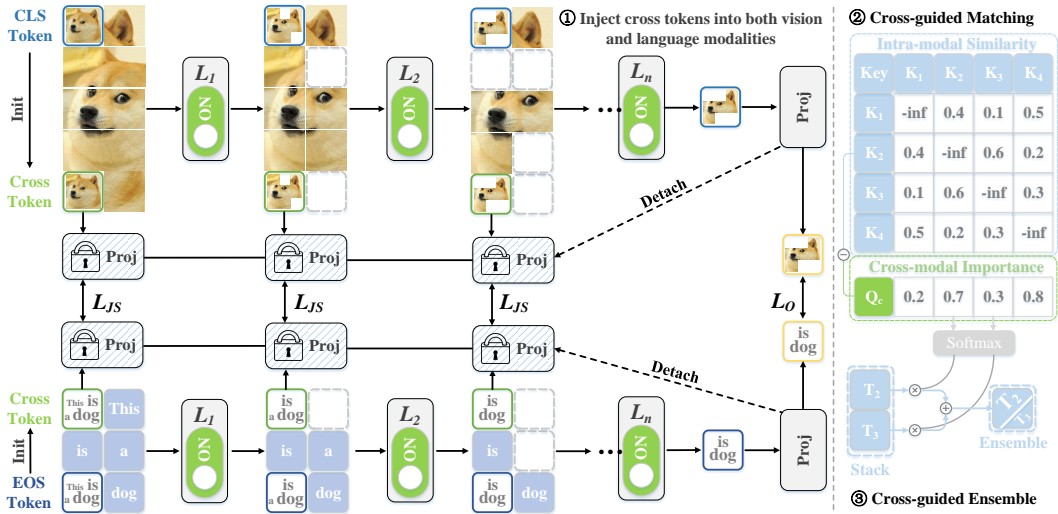

Figure 2: Diagram of introducing and exploiting cross-model guidance for vision-language Transformers. (1) Cross tokens are inserted into each layer and modality (the extra number of parameters is negligible, which is merely 0.01% of the total number); Cross tokens learn cross-modal information by closing the after-projection distance between cross tokens of different modalities in the same layer; The projection layers are detached from the last projection layer of the model, which does not introduce new learnable parameters and are not used during inference, having any impact on the inference efficiency of the model; The switches indicates it is free to choose whether to reduce tokens in different modalities and layers. (2) After learning cross-modal information, cross-tokens provide cross-modal importance as a metric to guide token matching. (3) The cross-modal importance also guides the token ensemble to produce a weighted summation of the stacked tokens as the ensemble results.

*CrossGET* addresses this challenge by decoupling the operation of inference on a modality and the ability to provide guidance for others, *i.e.*, in the previous example, modality2 can provide modality1 guidance without inference. As illustrated in Figure 2, this is achieved by injecting learnable cross tokens into each modality, driving them to learn cross-modal information from each other. When conducting inference within a modality, cross-tokens serve as agents for other modalities, providing cross-modal guidance on behalf of other modalities without inference.

**Token Matching with Cross-Modal Guidance**   Cross tokens can provide cross-modal importance as a metric to guide *complete-graph soft matching*. The cross-modal importance is computed as the cosine similarity between the query of the cross-token $\boldsymbol{T}_c \in \mathbb{R}^{1 \times d}$ where $d$ is the embedding size and the key of other tokens $\boldsymbol{T}_i \in \mathbb{R}^{1 \times d}$, $i \neq c$:

$$I_i = \frac{(\boldsymbol{T}_c \boldsymbol{W}^q)(\boldsymbol{T}_i \boldsymbol{W}^k)^\top}{\|\boldsymbol{T}_c \boldsymbol{W}^q\|_2 \|\boldsymbol{T}_i \boldsymbol{W}^k\|_2}, \tag{1}$$

where $\boldsymbol{W}^q, \boldsymbol{W}^k \in \mathbb{R}^{d \times d}$ are weights of query and key layers, respectively. $\| \cdot \|_2$ denotes L2-norm.

**Loss Function**   JS divergence (Menéndez et al., 1997), also known as a symmetrized KL divergence (Kullback & Leibler, 1951), between the projection of cross token $T_{cv}^i$ from vision and $T_{cl}^i$ from language in layer $i$ is (for modalities with different layers, order-preserving mappings can be used):

$$\mathcal{L}_{\mathcal{JS}}^i[(\boldsymbol{T}_{cv}^i \tilde{\boldsymbol{W}}^v) \| (\boldsymbol{T}_{cl}^i \tilde{\boldsymbol{W}}^l)] = \frac{1}{2} \left[ \mathcal{L}_{\mathcal{KL}}^i[(\boldsymbol{T}_{cv}^i \tilde{\boldsymbol{W}}^v) \| \boldsymbol{T}_m^i] + \mathcal{L}_{\mathcal{KL}}^i[(\boldsymbol{T}_{cl}^i \tilde{\boldsymbol{W}}^l) \| \boldsymbol{T}_m^i] \right], \tag{2}$$

$$\boldsymbol{T}_m^i = \frac{1}{2}(\boldsymbol{T}_{cv}^i \tilde{\boldsymbol{W}}^v + \boldsymbol{T}_{cl}^i \tilde{\boldsymbol{W}}^l), \tag{3}$$

where $\tilde{\boldsymbol{W}}^v$ and $\tilde{\boldsymbol{W}}^l$ are the detached weight of the final projection layer in vision modality and language modality, respectively. The detached implies $\mathcal{L}_{\mathcal{JS}}^i$ only produce gradients with respect to cross tokens $\boldsymbol{T}_{cv}^i$ and $\boldsymbol{T}_{cl}^i$ but not to projection layers, and the weight of projection layers $\boldsymbol{W}^v$ and $\boldsymbol{W}^l$ are updated only according to the gradients from original loss. $\mathcal{L}_{\mathcal{JS}}^i$ is added to drive cross tokens to learn cross-modal information from different modalities:

$$\mathcal{L} = \mathcal{L}_{\mathcal{O}} + 10^\alpha \sum_{i=0}^{L-1} \mathcal{L}_{\mathcal{JS}}^i, \tag{4}$$

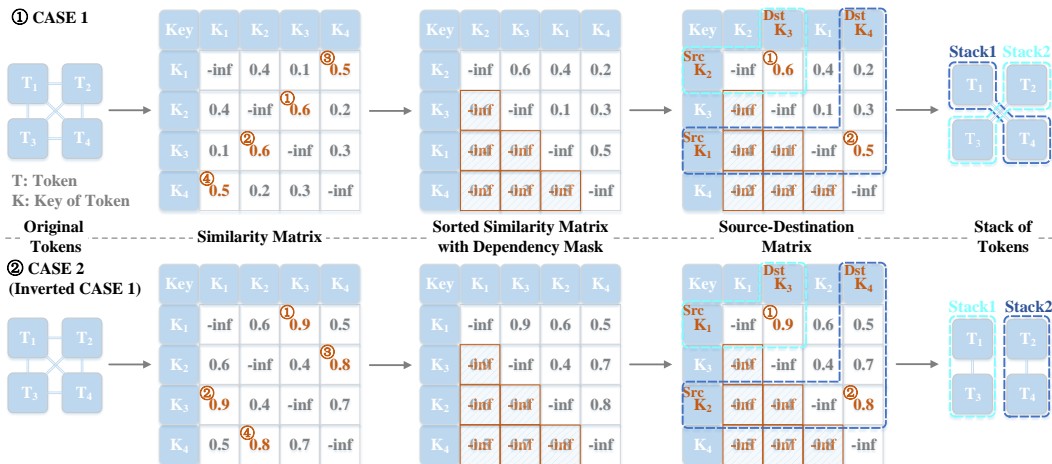

Figure 3: Illustration of complete-graph soft matching on two examples. Case2 is an inverted version of case1 in which the similarity between token pairs in case2 equals $(1-$ similarity of corresponding pairs in case1).

where $\mathcal{L}_{\mathcal{O}}$ is the original loss to learn a multimodal model, $\alpha$ is a hyperparameter to keep loss items to have the closest order of magnitude, and $L$ is the number of model layers, which means cross tokens are inserted into each layer of the model and $\mathcal{L}_{\mathcal{JS}}$ should be computed for each layer.

### 3.3 CROSS-GUIDED ENSEMBLE (CGE)

Benefit from the cross-modal importance metric that has already been used for guiding the token-matching, this design can be further improved by introducing cross-modal guidance into the ensemble, *i.e.*, using the softmax value of cross-modal importance to produce a weighted summation of the stacked tokens as the results of the ensemble:

$$T_i = \sum_{T_j \in S_i} \frac{e^{I_j}}{\sum_{T_k \in S_i} e^{I_k}} T_j, \tag{5}$$

where $S_i$ is the set of the stacked tokens, and $T_i$ is the corresponding ensembled token.

### 3.4 COMPLETE-GRAPH SOFT MATCHING (CGSM)

**Problem Formulation** The token matching aims to determine which tokens need to be combined. Suppose there are $N \in \mathbb{N}^+$ tokens in total, and $r \in \mathbb{N}^+$ $(r < N)$ tokens among them should be eliminated (*i.e.*, combined together with other tokens), then the problem of token matching can be formulated as a discrete optimization problem that is to find a set of feasible token pairs:

$$P = \{(T_i, T_j) \mid 0 \le i, j \le N, i \ne j\}, \quad |P| = r, \tag{6}$$

where $T_i$ denotes tokens $i$, and $|\cdot|$ denotes the size of the set, to maximize the objective function

$$S = \sum_{(T_i, T_j) \in P} \mathcal{D}(T_i, T_j), \tag{7}$$

where $\mathcal{D}$ is a function (*e.g.*, cosine similarity) that calculates the similarity between the key of the token $T_i$ and $T_j$. Appendix A.1 provides some examples to elaborate.

Clustering-based approaches can be used to obtain the solutions. However, they cannot be parallelized and are time-consuming. To make token matching parallelizable in mainstream deep learning frameworks (*e.g.*, Pytorch(Paszke et al., 2019)), an additional constraint should be satisfied:

$$T^S \cap T^D = \phi, \tag{8}$$

*i.e.*, the set of source tokens $T^S$ and destination tokens $T^D$ should be disjointed, where

$$T^S = \{T_i \mid (T_i, T_j) \in P\}, \quad |T^S| = r, \tag{9}$$

$$T^D = \{T_j \mid (T_i, T_j) \in P\}, \quad |T^D| \le r. \tag{10}$$

***Complete-Graph Soft Matching*** is designed as a non-iterative and approximate algorithm to ensure parallelizability and high efficiency. It enables each token to take into account the similarity with all other tokens as visualized in Figure 3 (Appendix A.2 provides an algorithm implementation):

- **Step 1**: For every two tokens, compute the cosine similarities of their keys to get the similarity matrix (self-similarities on the diagonal are set to $-\infty$ and ignored).
- **Step 2**: Sort rows and columns of the similarity matrix in descending order according to their maximum similarity to other tokens.
- **Step 3**: A lower triangle dependency mask is added upon the sorted similarity matrix to help disjoint set $T^S$ and $T^D$. It explicitly prioritizes the matching among tokens by the similarity value and ensures that source tokens with higher priority will not become targets for other source tokens with lower priority.
- **Step 4**: Pick $r$ rows with the maximum similarity to other tokens as the set of source tokens, and the columns corresponding to other tokens are the sets of destination tokens.
- **Step 5**: Each token in the source set should be stacked together with the token with the highest similarity in the destination set. All five steps are non-iterative and parallelizable.

As shown in Figure 3, *complete-graph soft matching* achieves optimal solutions on both case1 and case2. The modification in Step 4 is made for integrating *Cross-Guided Matching and Ensemble* to benefit from the cross-modal guidance (Appendix A.3 provides an algorithm implementation):

- **Step 4**: Pick $r$ rows with the maximum value of (maximum similarity to other tokens - cross-modal importance) as the set of source tokens, and the columns corresponding to other tokens are the sets of destination tokens.

Appendix A.4 and Appendix A.5 also provide further discussions on the sub-optimal cases of this method and analyses on the expectation of optimal matching probability, respectively.

## 4 EXPERIMENTS

We report the performance on modality-independent model CLIP (Radford et al., 2021) as well as modality-dependent models BLIP/BLIP2 (Li et al., 2022; 2023b), and mainstream tasks such as Image-Text Retrieval, Visual Reasoning, Image Captioning, and Visual Question Answering.

### 4.1 EXPERIMENTS WITH CLIP ON IMAGE-TEXT RETRIEVAL TASK

We conduct experiments on the CLIP model, and Flickr30K datasets (Young et al., 2014) with Karpathy split (Karpathy & Fei-Fei, 2015) of Image-Text Retrieval and Text-Image Retrieval task. The number of tokens is reduced to half with the same reduction number for each layer. For example, suppose one of the modalities of a 12-layer CLIP has 100 tokens as input, then $\lfloor \frac{100}{12} \rfloor = 8$ tokens will be eliminated from each layer so that the number of tokens left in the last layer is $100 - 12 \times 8 = 4$, and the total number of tokens across all layers is roughly reduced to half. If not specified, the number of tokens to be reduced in other experiments is also determined by this strategy.

**Comparison with Baselines**   Unless stated otherwise, all reported experimental results are after training. Table 1 demonstrates that *CrossGET* outperforms both the SOTA multimodal model pruning approach UPop (Shi et al., 2023), token reduction approach TRIPS (Jiang et al., 2022), and unimodal token reduction approach ToMe (Shi et al., 2023) without extra learnable parameter other than negligible cross tokens [2]. *CrossGET* is also evaluated against *ToMe* with adapter (Appendix C.11 provides details). It can also be observed that simply adding an extra learnable token to ToMe does not bring a notable improvement. In particular, the average of Recall@1 is significantly lower than *CrossGET*, which indicates that the improvement given by *cross-guided matching and ensemble* is mainly from learning cross-modal information instead of the increase of learnable tokens.

---

[2]For fairness of comparison, token reduction methods that require additional learnable parameters exceeding the level of several tokens are not taken into comparison (*e.g.*, simply adding a new linear projection layer with weight $W \in \mathbb{R}^{768 \times 768}$ already needs 768 times the number of our cross token's parameters $T_c \in \mathbb{R}^{1 \times 768}$)

Table 1: Accelerate CLIP on the Flickr30K dataset of the Image-Text Retrieval task. R: Recall. R@1, R@5 and R@10 are the higher the better. Experimental results are reported after training for all approaches. CrossGET$^{\blacktriangle}$ only uses *complete-graph soft matching (CGSM)* (Section 3.4), CrossGET$^{\blacklozenge}$ adds *cross-guided matching (CGM)* (Section 3.2) on $\blacktriangle$, and CrossGET$^{\star}$ further adds *cross-guided ensemble (CGE)* (Section 3.3) on $\blacklozenge$. Here *UPop* uses a larger CLIP as its original model, and therefore GFLOPs is higher.

| Approach | Image → Text | | | Text → Image | | | Avg. $\overline{\text{R@1}}$ | GFLOPs ↓ | Throughput ↑ |
|---|---|---|---|---|---|---|---|---|---|
| | R@1 | R@5 | R@10 | R@1 | R@5 | R@10 | | | |
| CLIP (Radford et al., 2021) | 92.1 | 99.1 | 99.7 | 79.3 | 95.7 | 98.0 | 85.7 | 20.6 | 255.2 |
| TRIPS (Jiang et al., 2022) | 90.4 | 98.9 | 99.5 | 76.8 | 94.4 | 97.2 | 83.6 | 16.4 | 316.9 |
| UPop (Shi et al., 2023) | 82.9 | 95.7 | 97.8 | 67.3 | 89.5 | 93.5 | 75.1 | 51.3 | - |
| Hourglass(Liang et al., 2022a) | 90.5 | 99.0 | 99.7 | 77.9 | 94.8 | 97.3 | 84.2 | 15.0 | 342.3 |
| DynamicViT(Rao et al., 2021) | 89.4 | 98.8 | 99.3 | 75.7 | 94.2 | 97.0 | 82.6 | 12.2 | 422.1 |
| EViT (Liang et al., 2022b) | 89.9 | 98.6 | 99.4 | 76.7 | 94.5 | 97.4 | 83.3 | 12.4 | 413.2 |
| ToMe (Bolya et al., 2023) | $90.8_{\downarrow1.3}$ | $99.2_{\uparrow0.1}$ | $99.5_{\downarrow0.2}$ | $78.1_{\downarrow1.2}$ | $95.3_{\downarrow0.4}$ | $97.7_{\downarrow0.3}$ | $84.5_{\downarrow1.2}$ | 11.8 | 417.4 |
| ToMe+Extra Token | $90.8_{\downarrow1.3}$ | $98.7_{\downarrow0.4}$ | $99.6_{\downarrow0.1}$ | $78.8_{\downarrow0.5}$ | $95.1_{\downarrow0.6}$ | $97.6_{\downarrow0.4}$ | $84.8_{\downarrow0.9}$ | 11.9 | 412.9 |
| ToMe+CGM & CGE | $91.5_{\downarrow0.6}$ | $99.0_{\downarrow0.1}$ | $99.6_{\downarrow0.1}$ | $78.6_{\downarrow0.7}$ | $95.4_{\downarrow0.3}$ | $97.8_{\downarrow0.2}$ | $85.1_{\downarrow0.6}$ | 11.9 | 409.9 |
| CrossGET$^{\blacktriangle}$ (CGSM) | $90.9_{\downarrow1.2}$ | $99.2_{\uparrow0.1}$ | $\mathbf{99.9}_{\uparrow0.2}$ | $79.1_{\downarrow0.2}$ | $95.1_{\downarrow0.6}$ | $97.6_{\downarrow0.4}$ | $85.0_{\downarrow0.7}$ | 11.9 | 408.9 |
| CrossGET$^{\blacklozenge}$ (CGM) | $\mathbf{92.1}_{\uparrow0.0}$ | $99.3_{\uparrow0.2}$ | $99.7_{\uparrow0.0}$ | $79.5_{\uparrow0.2}$ | $95.3_{\downarrow0.4}$ | $97.7_{\downarrow0.3}$ | $85.8_{\uparrow0.1}$ | 12.0 | 402.1 |
| CrossGET$^{\star}$ (Ours) | $\mathbf{92.1}_{\uparrow0.0}$ | $\mathbf{99.7}_{\uparrow0.6}$ | $\mathbf{99.8}_{\uparrow0.1}$ | $\mathbf{79.6}_{\uparrow0.3}$ | $\mathbf{95.7}_{\uparrow0.0}$ | $\mathbf{98.0}_{\uparrow0.0}$ | $\mathbf{85.9}_{\uparrow0.2}$ | $12.0_{\downarrow42\%}$ | $401.8_{\uparrow57\%}$ |

**Effect of individual components** As highlighted by grey in Table 1, *complete-graph soft matching* (CGSM) brings improvements on most of the metrics and a significant improvement on text-to-image retrieval (recall@1 increases from 78.1 to 79.1). Since the complete graph has more similarity of token pairs to compute than the bipartite graph, GFLOPs also slightly increases by 0.1. *Cross-guided matching* (CGM) brings further improvement on most metrics and a significant improvement on image-to-text retrieval (recall@1 increases from 90.9 to 92.1). Since cross tokens interact with other tokens during the forward, GFLOPs again slightly increases by 0.1. *Cross-guided ensemble* brings final improvement on all metrics with negligible extra GFLOPs. Moreover, consistent improvements can also be observed when *Cross-Guided Matching and Ensemble* is applied to *ToMe*. Compared to the original CLIP, *CrossGET* achieves the same image-to-text recall@1 and 0.3 higher text-to-image recall@1 while saving 42% GFLOPs and improving throughput by 57%.

## 4.2 EXPERIMENTS WITH BLIP ON THE VISUAL REASONING TASK

Table 2: Accelerate BLIP on the NLVR2 dataset of the Vision Reasoning task. BLIP is the original model for all acceleration approaches.

| Approach | Dev Acc | Test Acc | GFLOPs | Throughput |
|---|---|---|---|---|
| BLIP (Li et al., 2022) | 82.3 | 83.4 | 132.5 | 39.8 |
| UPop (Shi et al., 2023) | $80.3_{\downarrow2.0}$ | $81.1_{\downarrow2.3}$ | 89.4 | - |
| ToMe (Bolya et al., 2023) | $81.7_{\downarrow0.6}$ | $82.2_{\downarrow1.2}$ | 59.0 | 81.9 |
| CrossGET$^{\blacktriangle}$ (CGSM) | $82.2_{\downarrow0.1}$ | $82.6_{\downarrow0.8}$ | 60.8 | 77.7 |
| CrossGET$^{\star}$ (Ours) | $82.1_{\downarrow0.2}$ | $\mathbf{83.2}_{\downarrow0.2}$ | $61.1_{\downarrow57\%}$ | $76.8_{\uparrow93\%}$ |

Table 2 shows *CrossGET* also achieves very competitive performance on the BLIP model and NLVR2 dataset of a vision reasoning task that requires predicting whether a given sentence can describe a pair of given images. Compared to the original BLIP, *CrossGET* gets only 0.2 lower accuracies on the dev set and test set while saving 57% GFLOPs and improving throughput by 93%.

## 4.3 EXPERIMENTS AT DIFFERENT REDUCTION RATIOS

Figure 4 illustrates experimental results at various reduction ratios under three different settings: (1) Comparisons without training[3] (left subfigure); (2) Comparisons with training (upper-right subfigure). (3) Re-evaluate a trained model (50% token reduced) under other token reduction ratios without training (lower-right subfigure). These subfigures demonstrate that *CrossGET* achieves superior Pareto frontiers in all three situations. Appendix C.15, C.16, and C.17 provide detailed data. Besides, the original ToMe method does not require training, and the comparison with it is illustrated in the small plot at the lower right corner of the left subfigure.

---

[3]The only part of *CrossGET* that requires training is learning cross tokens. However, they are initialized with informative features (see Appendix C.5) and already contain representative information even though they are not trained. Therefore, *CrossGET* can also be used without training (certainly worse than with training).

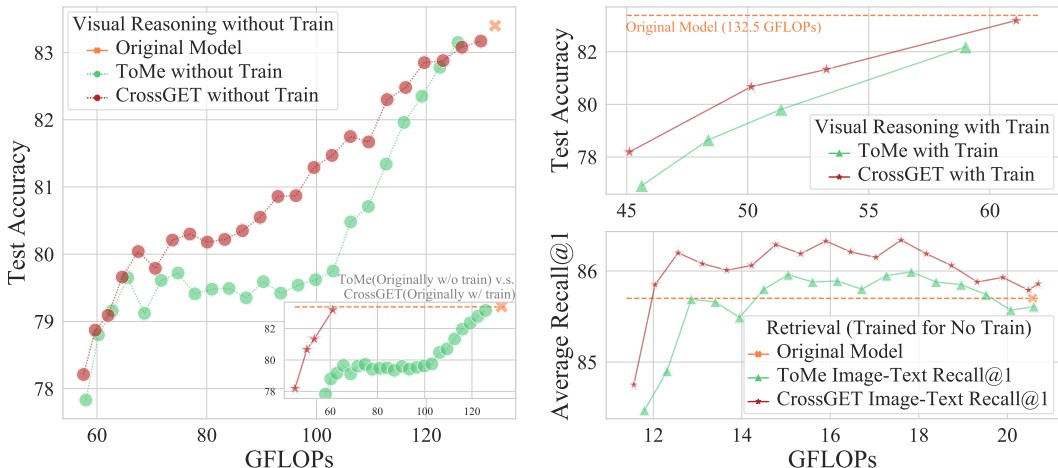

Figure 4: Peformance-Cost tradeoffs in three situations: 1) The left subfigure illustrates the tradeoff for BLIP on the NVLR2 dataset of the Visual Reasoning task without training. 2) The upper-right subfigure illustrates the tradeoff for BLIP on the NVLR2 dataset of the Visual Reasoning task with training. 3) The lower-right subfigure illustrates the tradeoff for CLIP on the Flickr30K dataset of the Image-Text Retrieval task that we train a model with 50% token reduced and then re-evaluated under other token reduction ratios without training.

## 4.4 EXPERIMENTS WITH BLIP ON THE IMAGE CAPTIONING TASK

On auto-regressive models performing cross-modal interactions at each layer and forward via Cross-Attentions, such as the BLIP-Captioning (Li et al., 2022) model, *CrossGET* achieves higher speedups. As shown in Table 3, reducing the total tokens by half for the generation brings 86% saving of GFLOPs and improving 330% throughput.

We also conduct experiments on the NoCaps (Agrawal et al., 2019) datasets of the Novel Object Caption task, and the model accelerated by *CrossGET* again achieves superior performances.

Table 3: Accelerate BLIP on the COCO Caption dataset of the Image Caption task. The suffix -F denotes GFLOPs and throughput for the forward, while -G denotes GFLOPs and throughput for the generation.

| Approach | CIDEr | SPICE | GFLOPs-F | Throughput-F | GFLOPs-G | Throughput-G |
|---|---|---|---|---|---|---|
| BLIP (Li et al., 2022) | 133.3 | 23.8 | 65.7 | 106.4 | 330.7 | 17.2 |
| UPop (Shi et al., 2023) | $128.9_{\downarrow 4.4}$ | $23.3_{\downarrow 0.5}$ | 39.8 | - | - | - |
| ToMe (Bolya et al., 2023) | $130.3_{\downarrow 3.0}$ | $23.3_{\downarrow 0.5}$ | 29.2 | 209.3 | 43.8 | 77.7 |
| CrossGET (Ours) | $\mathbf{131.6}_{\downarrow 1.7}$ | $\mathbf{23.8}_{\uparrow 0.0}$ | $30.1_{\downarrow 54\%}$ | $183.5_{\uparrow 72\%}$ | $46.7_{\downarrow 86\%}$ | $73.9_{\uparrow 330\%}$ |

Table 4: Accelerate BLIP on the NoCaps dataset of the Novel Object Caption task. All metrics are the higher the better, and the evaluation uses the same model finetuned on the COCO Caption dataset as in Table 3, and therefore the GFLOPs and throughput of models are the same as in Table 3.

| Approach | in-domain | | near-domain | | out-domain | | **entire** | |
|---|---|---|---|---|---|---|---|---|
| | CIDEr | SPICE | CIDEr | SPICE | CIDEr | SPICE | CIDEr | SPICE |
| BLIP (Li et al., 2022) | 111.9 | 14.9 | 108.8 | 14.8 | 112.1 | 14.2 | 109.9 | 14.7 |
| ToMe (Bolya et al., 2023) | $107.9_{\downarrow 4.0}$ | $14.8_{\downarrow 0.1}$ | $105.1_{\downarrow 3.7}$ | $14.4_{\downarrow 0.4}$ | $106.4_{\downarrow 5.7}$ | $14.1_{\downarrow 0.1}$ | $105.7_{\downarrow 4.2}$ | $14.4_{\downarrow 0.3}$ |
| CrossGET (Ours) | $\mathbf{113.2}_{\uparrow 1.3}$ | $\mathbf{15.1}_{\uparrow 0.2}$ | $107.2_{\downarrow 1.6}$ | $\mathbf{14.6}_{\downarrow 0.2}$ | $107.4_{\downarrow 4.7}$ | $14.1_{\downarrow 0.1}$ | $\mathbf{108.1}_{\downarrow 1.8}$ | $\mathbf{14.6}_{\downarrow 0.1}$ |

## 4.5 EXPERIMENTS WITH BLIP ON THE VISUAL QUESTION ANSWER TASK

We conduct experiments on the BLIP model (Li et al., 2022) and the test-dev set of the VQA2.0 dataset (Goyal et al., 2017). Table 5 demonstrates that *CrossGET* can also considerably save computational cost and improve throughput for the Visual Question Answer task. For example, compared to the original model, *CrossGET* gets only 0.4 lower overall accuracy on all three types of questions while

Table 5: Accelerate BLIP on the VQA2.0 dataset of the Visual Question Answer task. "yes/no", "number", "other", and "overall" denote accuracy on the corresponding types of questions. These four metrics are the higher the better. The suffix -F denotes GFLOPs and throughput(tput.) for the forward that a single image may be accompanied by multiple questions and answers during training, while -T denotes GFLOPs and throughput(tput.) for the test that a single image is accompanied by only one question and answer.

| Approach | yes/no | number | other | **overall** | GFLOPs-F | Tput.-F | GFLOPs-T | Tput.-T |
|---|---|---|---|---|---|---|---|---|
| BLIP (Li et al., 2022) | 92.6 | 60.6 | 68.3 | 77.4 | 186.1 | 67.2 | 106.8 | 53.0 |
| UPop (Shi et al., 2023) | - | - | - | $76.3_{\downarrow 1.1}$ | 109.4 | - | - | - |
| ToMe (Bolya et al., 2023) | $92.1_{\downarrow 0.5}$ | $59.3_{\downarrow 1.3}$ | $67.1_{\downarrow 1.2}$ | $76.5_{\downarrow 0.9}$ | 119.0 | 141.1 | 46.7 | 90.1 |
| CrossGET (Ours) | $\mathbf{92.4}_{\downarrow 0.2}$ | $\mathbf{59.7}_{\downarrow 0.9}$ | $\mathbf{67.7}_{\downarrow 0.6}$ | $\mathbf{77.0}_{\downarrow 0.4}$ | $124.5_{\downarrow 33\%}$ | $120.4_{\uparrow 79\%}$ | $49.0_{\downarrow 54\%}$ | $81.3_{\uparrow 53\%}$ |

Table 6: Accelerate multimodal LLM BLIP2-OPT6.7B (Li et al., 2023b) on the COCO Caption dataset of the Image Caption task. The suffix -F denotes GFLOPs and throughput for the forward, while -G denotes GFLOPs and throughput for the generation. Experimental results on BLIP2-OPT2.7B are provided in Appendix C.13.

| Approach | Tuning | CIDEr | BLEU@4 | GFLOPs-F | Tput.-F | GFLOPs-G | Tput.-G |
|---|---|---|---|---|---|---|---|
| BLIP2-OPT6.7B | - | 144.5 | 42.5 | 1042.6 | 47.4 | 2461.1 | 16.2 |
| | w/o tuning | 144.7 | **42.4** | 957.6 | - | 2342.7 | - |
| | w/o tuning | 144.3 | 42.3 | 868.1 | - | 2086.7 | - |
| ToMe | w/o tuning | 142.4 | 41.9 | 780.7 | - | 2232.4 | - |
| (Bolya et al., 2023) | w/o tuning | 135.5 | 40.1 | 695.1 | - | 2046.9 | - |
| | w/ tuning | $141.7_{\downarrow 2.8}$ | $41.4_{\downarrow 1.1}$ | 544.8 | 92.6 | 1510.0 | 21.5 |
| | w/o tuning | 144.5 | 42.3 | 973.8 | - | 2392.3 | - |
| | w/o tuning | 144.6 | **42.4** | 881.1 | - | 2266.2 | - |
| CrossGET(Ours) | w/o tuning | 143.3 | **42.1** | 790.9 | - | 2176.1 | - |
| | w/o tuning | 137.5 | **40.6** | 703.4 | - | 2121.8 | - |
| | w/ tuning | $\mathbf{143.1}_{\downarrow 1.4}$ | $\mathbf{42.0}_{\downarrow 0.5}$ | $558.2_{\downarrow 49\%}$ | $91.0_{\uparrow 92\%}$ | $1583.2_{\downarrow 36\%}$ | $21.6_{\uparrow 33\%}$ |

saving 33% GFLOPs and improving throughput by 79% for the multiple-question scenario, and saving 54% GFLOPs and improving throughput by 53% for the single-question scenario.

## 4.6 EXPERIMENTS WITH BLIP2 ON THE IMAGE CAPTIONING TASK

We also apply CrossGET on BLIP2 (Li et al., 2023b). We follow the original strategy of BLIP2 that tunes ViT and Q-Former(a Bert) while freezing the LLM. Experimental results given by tuning and without tuning are provided. Table 6 demonstrates that CrossGET also achieves consistently promising performance on multimodal LLMs. Besides, compared with BLIP, the language branch of BLIP2 receives fewer tokens from the vision branch and thus yields less generation speedup.

## 4.7 ADDITIONAL EXPERIMENTS AND ABLATION STUDY

Please refer to Appendix C for additional experiments and ablation study, including ablation study on training hyperparameters (Appendix C.2), generic (Appendix C.3 ∼ C.7) and specific (Appendix C.8 ∼ C.9) design choices, comparison experiments with more baselines (Appendix C.10 ∼ C.11), more results on CoOp (Appendix C.12) and BLIP2 (Appendix C.13 ∼ C.14), and results at different reduction ratios (Appendix C.15 ∼ C.17).

## 5 CONCLUSION

In this work, we introduced *CrossGET*, a universal token ensemble framework tailored for the acceleration of vision-language Transformers. *CrossGET* utilizes cross-modal guidance to make informed decisions on token selection and ensemble, thereby optimizing the performance-to-computation trade-off. Notably, our token-matching methodology is grounded on a complete graph, ensuring superior token-matching reliability in comparison to the prevalent bipartite graph-based approaches. In summary, *CrossGET* provides a favorable performance-efficiency trade-off and demonstrates robust applicability, as evidenced through extensive empirical evaluations on a multitude of vision-language tasks, datasets, and model architectures.

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

# A  SUPPLEMENTARY METHODOLOGY DETAILS

## A.1  EXAMPLES FOR DEMONSTRATING TOKEN MATCHING

**Optimal Objective Function and Solution**    For example, when the number of tokens in total is $N = 4$, and the number of tokens to be reduced is $r = 2$, by verifying and comparing all possible token-matching results, the optimal objective function for case1 in Figure 3 can be obtained:

$$S_1^* = \mathcal{D}(\boldsymbol{T}_1, \boldsymbol{T}_4) + \mathcal{D}(\boldsymbol{T}_2, \boldsymbol{T}_3) = 0.5 + 0.6 = 1.1, \tag{11}$$

and the corresponding optimal solution for token matching can be determined as

$$\boldsymbol{P}_1^* = \{(\boldsymbol{T}_1, \boldsymbol{T}_4), (\boldsymbol{T}_2, \boldsymbol{T}_3)\}. \tag{12}$$

Similarly, the optimal objective function for case2 (inverted case1) in Figure 3 is

$$S_2^* = \mathcal{D}(\boldsymbol{T}_1, \boldsymbol{T}_3) + \mathcal{D}(\boldsymbol{T}_2, \boldsymbol{T}_4) = 0.9 + 0.8 = 1.7 \tag{13}$$

and the corresponding optimal solution for token matching is

$$\boldsymbol{P}_2^* = \{(\boldsymbol{T}_1, \boldsymbol{T}_3), (\boldsymbol{T}_2, \boldsymbol{T}_4)\}. \tag{14}$$

**Revisiting Bipartite Soft Matching**    ToMe (Bolya et al., 2023) suggests a non-iterative *bipartite soft matching* ensure parallelizability, which divides tokens into two disjoint sets alternately, for each token in the first set calculates the maximum similarity from it to each token in the other set, and the token pairs with the highest similarities will be merged.

Take case1 in Figure 3 as an example, tokens are firstly divided into $\{\boldsymbol{T}_1, \boldsymbol{T}_3\}$ and $\{\boldsymbol{T}_2, \boldsymbol{T}_4\}$, then the maximum similarity from $\boldsymbol{T}_1$ to $\{\boldsymbol{T}_2, \boldsymbol{T}_4\}$ is $\mathcal{D}(\boldsymbol{T}_1, \boldsymbol{T}_4) = 0.5$ and from $\boldsymbol{T}_3$ to $\{\boldsymbol{T}_2, \boldsymbol{T}_4\}$ is $\mathcal{D}(\boldsymbol{T}_3, \boldsymbol{T}_2) = 0.6$. Therefore, the optimal objective function in Eq.11 and optimal solution in Eq.12 are achieved:

$$S_1^B = \mathcal{D}(\boldsymbol{T}_1, \boldsymbol{T}_4) + \mathcal{D}(\boldsymbol{T}_2, \boldsymbol{T}_3) = S_1^*, \quad \boldsymbol{P}_1^B = \{(\boldsymbol{T}_1, \boldsymbol{T}_4), (\boldsymbol{T}_2, \boldsymbol{T}_3)\} = \boldsymbol{P}_1^* \tag{15}$$

However, for case2 (inverted case1), *bipartite soft matching* leads to a worse solution: tokens are firstly divided into $\{\boldsymbol{T}_1, \boldsymbol{T}_3\}$ and $\{\boldsymbol{T}_2, \boldsymbol{T}_4\}$, then the maximum similarity from $\boldsymbol{T}_1$ to $\{\boldsymbol{T}_2, \boldsymbol{T}_4\}$ is $\mathcal{D}(\boldsymbol{T}_1, \boldsymbol{T}_2) = 0.6$ and from $\boldsymbol{T}_3$ to $\{\boldsymbol{T}_2, \boldsymbol{T}_4\}$ is $\mathcal{D}(\boldsymbol{T}_3, \boldsymbol{T}_4) = 0.7$. Therefore, the optimal objective function in Eq.13 and optimal solution in Eq.14 are not achieved:

$$S_2^B = \mathcal{D}(\boldsymbol{T}_1, \boldsymbol{T}_2) + \mathcal{D}(\boldsymbol{T}_3, \boldsymbol{T}_4) < S_2^*, \quad \boldsymbol{P}_2^B = \{(\boldsymbol{T}_1, \boldsymbol{T}_2), (\boldsymbol{T}_3, \boldsymbol{T}_4)\} \neq \boldsymbol{P}_2^* \tag{16}$$

This is attributed to the design of *bipartite soft matching* that for the convenience of ensuring parallelizability, each token only takes into account the similarity with half but not all other tokens, and the method degrades when tokens with high similarity are not divided into different sets.

**Shifting to Complete-Graph Soft Matching**    An approximate algorithm *complete-graph soft matching* is proposed to tackle the above challenge. It enables each token to take into account the similarity with all other tokens while avoiding introducing iterative and non-parallelizable operations.

Take case2 in Figure 3 as an example, all tokens $\boldsymbol{T} = \{\boldsymbol{T}_1, \boldsymbol{T}_2, \boldsymbol{T}_3, \boldsymbol{T}_4\}$ are sorted in descending order according to their maximum similarity to other tokens: $\boldsymbol{T}' = \{\boldsymbol{T}_1, \boldsymbol{T}_3, \boldsymbol{T}_2, \boldsymbol{T}_4\}$. After adding the dependency mask, the maximum similarity from top priority source token candidate $\boldsymbol{T}_1$ to its destination tokens $\{\boldsymbol{T}_3, \boldsymbol{T}_2, \boldsymbol{T}_4\}$ is $\mathcal{D}(\boldsymbol{T}_1, \boldsymbol{T}_3) = 0.9$, from second priority source token candidate $\boldsymbol{T}_3$ to its destination tokens $\{\boldsymbol{T}_2, \boldsymbol{T}_4\}$ is $\mathcal{D}(\boldsymbol{T}_3, \boldsymbol{T}_4) = 0.7$, and from third priority source token candidate $\boldsymbol{T}_2$ to its destination token $\boldsymbol{T}_4$ is $\mathcal{D}(\boldsymbol{T}_2, \boldsymbol{T}_4) = 0.8$. The source token candidates among them corresponding to the two largest similarities are selected as the source token set $\boldsymbol{T}_s = \{\boldsymbol{T}_1, \boldsymbol{T}_2\}$ while remaining tokens form the destination token set $\boldsymbol{T}_D = \{\boldsymbol{T}_3, \boldsymbol{T}_4\}$. Then the maximum similarity from $\boldsymbol{T}_1$ to $\{\boldsymbol{T}_3, \boldsymbol{T}_4\}$ is $\mathcal{D}(\boldsymbol{T}_1, \boldsymbol{T}_3) = 0.9$ and from $\boldsymbol{T}_2$ to $\{\boldsymbol{T}_3, \boldsymbol{T}_4\}$ is $\mathcal{D}(\boldsymbol{T}_2, \boldsymbol{T}_4) = 0.8$. Therefore, the optimal objective function in Eq.13 and optimal solution in Eq.14 are achieved:

$$S_2^C = \mathcal{D}(\boldsymbol{T}_1, \boldsymbol{T}_3) + \mathcal{D}(\boldsymbol{T}_2, \boldsymbol{T}_4) = S_2^*, \quad \boldsymbol{P}_2^C = \{(\boldsymbol{T}_1, \boldsymbol{T}_3), (\boldsymbol{T}_2, \boldsymbol{T}_4)\} = \boldsymbol{P}_2^*. \tag{17}$$

Similarly, it can also be verified that *complete-graph soft matching* achieves the optimal objective function in Eq.11 and optimal solution in Eq.12 for case1 in Figure 3.

## A.2 ALGORITHM IMPLEMENTATION OF COMPLETE-GRAPH SOFT MATCHING

---

**Algorithm 1:** Complete-Graph Soft Matching

---

**Input:** Number of tokens $N$, number of tokens to be reduced $r$, original tokens $\boldsymbol{T} = \{\boldsymbol{T}_i\}_{i=1}^N$
and their corresponding keys $\boldsymbol{K} = \{\boldsymbol{K}_i\}_{i=1}^N$ where $|\boldsymbol{T}| = |\boldsymbol{K}| = N$

**Output:** Reduced tokens $\boldsymbol{T}^\star = \{\boldsymbol{T}_i^\star\}_{i=1}^{N-r}$ where $|\boldsymbol{T}^\star| = N - r$

1 # Step1: Calculate the cosine distance $\boldsymbol{D}_{ij}$ between the keys of tokens

2 $\boldsymbol{D} = \frac{\boldsymbol{K}\boldsymbol{K}^\top}{\|\boldsymbol{K}\|_2^2} + \mathrm{diag}(\underbrace{-\infty, -\infty, \cdots, -\infty}_{N}), \quad \boldsymbol{D} \in \mathbb{R}^{N \times N}, \ \mathrm{diag} : \mathbb{R}^N \to \mathbb{R}^{N \times N}$

3 # Step2: Descendingly sort similarity matrix $\boldsymbol{D}$ by maximum similarity

4 $\boldsymbol{A}^S = \mathrm{argsort}(\max_{1 \leq j \leq N} \boldsymbol{D}_{ij}) \in \mathbb{R}^N, \quad \boldsymbol{A}^D = \mathrm{argsort}(\max_{1 \leq i \leq N} \boldsymbol{D}_{ij}) \in \mathbb{R}^N$

5 $\boldsymbol{D}^\star = \mathrm{sort}_\mathrm{d}(\mathrm{sort}_\mathrm{s}(\boldsymbol{D}, \boldsymbol{A}^S), \boldsymbol{A}^D), \ \mathrm{sort}_\mathrm{s} : \boldsymbol{D}_{ij}^\star \leftarrow \boldsymbol{D}_{\boldsymbol{A}_i^S j}, \ \mathrm{sort}_\mathrm{d} : \boldsymbol{D}_{ij}^\star \leftarrow \boldsymbol{D}_{i\boldsymbol{A}_j^D}$

6 # Step3: Add a lower triangle dependency mask $\boldsymbol{M}$

7 $\boldsymbol{D}^\star = \boldsymbol{D}^\star + \boldsymbol{M}, \quad \boldsymbol{M}_{ij} = \begin{cases} -\infty & \text{for } i \geq j \\ 0 & \text{for } i < j \end{cases}$

8 # Step4: Pick source tokens $\boldsymbol{T}^S$ and destination tokens $\boldsymbol{T}^D$ by similarity

9 $\boldsymbol{A} = \mathrm{argsort}(\max_{1 \leq j \leq N} \boldsymbol{D}_{ij}^\star) \in \mathbb{R}^N, \quad \boldsymbol{A}^S = (\boldsymbol{A}_i)_{1 \leq i \leq r} \in \mathbb{R}^r, \quad \boldsymbol{T}^S = \{\boldsymbol{T}_i | i \in \boldsymbol{A}^S\}$

10 $\boldsymbol{A} = \mathrm{argmax}_{j \in (\{k\}_{k=1}^N \setminus \boldsymbol{A}^S)} \boldsymbol{D}_{ij}^\star \in \mathbb{R}^N, \quad \boldsymbol{A}^D = (\boldsymbol{A}_i)_{i \in \boldsymbol{A}^s} \in \mathbb{R}^r, \quad \boldsymbol{T}^D = \{\boldsymbol{T}_i | i \in \boldsymbol{A}^D\}$

11 # Step5: Average source and corresponding destination tokens

12 **return** $\boldsymbol{T}^\star = [\boldsymbol{T} \setminus (\boldsymbol{T}^S \cup \boldsymbol{T}^D)] \cup \{\frac{1}{2}(\boldsymbol{T}_i^S + \boldsymbol{T}_i^D)\}_{i=1}^r$

---

Algorithm 1 is the detailed implementation of the proposed *complete-graph soft matching*. The Step1 $\sim$ 5 in the comments of Algorithm 1 correspond to the Step1 $\sim$ 5 described in Section 3.4 of the main text. Moreover, Algorithm 1 is similar to ToMe (Bolya et al., 2023) in terms of parallelizability that there are no sequential loops, and therefore data can be processed in parallel within each step by parallelizable operations (such as *bmm*, *matmul*, *scatter* and *gather* in Pytorch (Paszke et al., 2019)).

## A.3 ALGORITHM IMPLEMENTATION OF CROSS-GUIDED MATCHING AND ENSEMBLE

---

**Algorithm 2:** Cross-Guided Matching and Ensemble (improvements upon Algorithm 1)

---

**Input:** Same inputs as Algorithm 1, plus query of the cross token $\boldsymbol{Q}$

**Output:** Same as Algorithm 1

1 # Step1~3: Same as Algorithm 1

2 # Step4: Pick tokens $\boldsymbol{T}^S$ and $\boldsymbol{T}^D$ by similarity and importance

3 $\boldsymbol{I} = \frac{\boldsymbol{K}\boldsymbol{Q}^\top}{\|\boldsymbol{K}\|_2 \|\boldsymbol{Q}\|_2} \in \mathbb{R}^N$

4 $\boldsymbol{A} = \mathrm{argsort}(\max_{1 \leq j \leq N} \boldsymbol{D}_{ij}^\star - \boldsymbol{I}) \in \mathbb{R}^N, \quad \boldsymbol{A}^S = (\boldsymbol{A}_i)_{1 \leq i \leq r} \in \mathbb{R}^r, \quad \boldsymbol{T}^S = \{\boldsymbol{T}_i | i \in \boldsymbol{A}^S\}$

5 $\boldsymbol{A} = \mathrm{argmax}_{j \in (\{k\}_{k=1}^N \setminus \boldsymbol{A}^S)} \boldsymbol{D}_{ij}^\star \in \mathbb{R}^N, \quad \boldsymbol{A}^D = (\boldsymbol{A}_i)_{i \in \boldsymbol{A}^s} \in \mathbb{R}^r, \quad \boldsymbol{T}^D = \{\boldsymbol{T}_i | i \in \boldsymbol{A}^D\}$

6 # Step5: Sum weighted source and corresponding destination tokens

7 $\boldsymbol{P} = \{(\boldsymbol{T}_i^S, \boldsymbol{T}_i^D)\}_{i=1}^r, \quad \boldsymbol{W} = \{\mathrm{softmax}(\boldsymbol{I}_i, \boldsymbol{I}_j) | (\boldsymbol{T}_i, \boldsymbol{T}_j) \in \boldsymbol{P}\}$

8 **return** $\boldsymbol{T}^\star = [\boldsymbol{T} \setminus (\boldsymbol{T}^S \cup \boldsymbol{T}^D)] \cup \{\sum_{j=1}^{|\boldsymbol{W}_i|} \boldsymbol{W}_{ij} \boldsymbol{P}_{ij}\}_{i=1}^r$

---

Algorithm 2 demonstrates how to improve *complete-graph soft matching* by adding *cross-guided matching and ensemble* upon it. It is worth noting that line7 $\sim$ 8 in Algorithm 2 does not imply that only two tokens are in each stack of tokens to be ensembled. This is because different source tokens in $\boldsymbol{T}^S$ may have the same destination token in $\boldsymbol{T}^D$, which implies that the size of the stack is allowed to be larger than two (in this case, the procedure of ensembling stacks with the different number of

tokens can still be implemented by parallelizable operations such as *scatter_add* in Pytorch(Paszke et al., 2019)).

## A.4  SUB-OPTIMAL CASES FOR COMPLETE-GRAPH SOFT MATCHING

Section 3.4 has already shown the cases that *complete-graph soft matching* achieves optimal matching, and here we provide more analyses on the sub-optimal cases of *complete-graph soft matching*.

The main sub-optimal cases come from the trade-off between parallelizability and matching accuracy. To achieve parallelizability, the set of source token $\boldsymbol{T}^S$ and destination tokens $\boldsymbol{T}^D$ have to be disjoint:

$$\boldsymbol{T}^S \cap \boldsymbol{T}^D = \phi. \tag{18}$$

Otherwise, consider

$$\boldsymbol{T}^S \cap \boldsymbol{T}^D = \{\boldsymbol{T}_x\} \neq \phi, \quad 1 \leq x \leq N \tag{19}$$

where $N$ is the number of the original tokens, then

$$(\exists \boldsymbol{T}_i \in \boldsymbol{T}^S \text{ s.t. } (\boldsymbol{T}_i, \boldsymbol{T}_x) \in \boldsymbol{P}) \wedge (\exists \boldsymbol{T}_j \in \boldsymbol{T}^D \text{ s.t. } (\boldsymbol{T}_x, \boldsymbol{T}_j) \in \boldsymbol{P}) \tag{20}$$

where $\boldsymbol{P} = \{(\boldsymbol{T}_i^S, \boldsymbol{T}_i^D)\}_{i=1}^r$ is the set of the paired tokens to be ensembled, is true. However, there is a computational dependency between merging $\boldsymbol{T}_i$ into $\boldsymbol{T}_x$ and merging $\boldsymbol{T}_x$ into $\boldsymbol{T}_j$. The two operations of the merging require iterations and therefore cannot be parallelized.

$\boldsymbol{T}^S$ and $\boldsymbol{T}^D$ are disjoint (*i.e.*, Eq.18 holds) is equivalent to the constraint

$$\forall \boldsymbol{T}_i \in \boldsymbol{T}^S, \boldsymbol{T}_i \notin \boldsymbol{T}^D \tag{21}$$

is satisfied. In the Step1 of the Algorithm 1, computation is conducted on a complete graph. Therefore $\boldsymbol{T}^S$ and $\boldsymbol{T}^D$ are joint, and constraint 21 does not been satisfied. In Step3, the added lower triangle dependency mask ensures that source tokens with higher priority (*i.e.*, whose keys have higher maximum cosine similarity to keys of other tokens) will not become targets for other source tokens with lower priority, *i.e.*, a relaxed constraint

$$\forall \boldsymbol{T}_i \in \boldsymbol{T}^S, \boldsymbol{T}_i \notin (\boldsymbol{T}_j^D)_{i \leq j \leq N} \tag{22}$$

is satisfied. However, the unsatisfied part of the constraint 21

$$\forall \boldsymbol{T}_i \in \boldsymbol{T}^S, \boldsymbol{T}_i \notin (\boldsymbol{T}_j^D)_{1 \leq j < i} \tag{23}$$

indicates that source tokens with low priority may still become targets for other source tokens with high priority. To further satisfy constraint 23, the line10 of Step4 in Algorithm 1 explicitly removes all elements of the source token set from the set of all tokens to construct the set of the destination tokens. And the sub-optimal cases for *complete-graph soft matching* arise when

$$\underset{j \in (\{k\}_{k=1}^N \setminus \boldsymbol{A}^S)}{\operatorname{argmax}} \boldsymbol{D}_{ij}^\star \neq \underset{j \in \{k\}_{k=1}^N}{\operatorname{argmax}} \boldsymbol{D}_{ij}^\star, \tag{24}$$

which indicates a source token may exist whose closest destination token in $\boldsymbol{T}^D$ happens to be another source token in $\boldsymbol{T}^S$. For parallelizability, this destination token is removed from $\boldsymbol{T}^D$, resulting in the source token can only match the second closest destination token in the set of reduced $\boldsymbol{T}^D$.

## A.5  EXPECTATION OF OPTIMAL MATCHING PROBABILITY AND COMPLEXITY ANALYSIS

**Expectation of Optimal Matching Probability**    For a token $\boldsymbol{T}_i \in \boldsymbol{T}$, assume that any other token $\boldsymbol{T}_j \in \boldsymbol{T} \setminus \{\boldsymbol{T}_i\}$ has the same probability of being its optimal destination token, *i.e.*

$$\forall 1 \leq j \leq N, \quad p((\boldsymbol{K}_i, \boldsymbol{K}_j) = \underset{\substack{1 \leq k \leq N \\ k \neq i}}{\operatorname{argmax}} s(\boldsymbol{K}_i, \boldsymbol{K}_k)) = \frac{1}{N-1} \tag{25}$$

where $s(x, y)$ is a function that calculates cosine similarity between $x$ and $y$.

For *complete-graph soft matching*, in layer $l$ ($1 \leq l \leq L$), supoose $\boldsymbol{X} \sim p(x)$ is a discrete random variable about whether a token from $\boldsymbol{T}^S$ ($|\boldsymbol{T}^S| = r$) can find its optimal destination token in $\boldsymbol{T}^D$

$(|\boldsymbol{T}^D| = N - lr)$, and the probability distribution $p(x)$ is:

$$p(\boldsymbol{X} = \text{can}) = \sum_1^{|\boldsymbol{T}^D|} p((\boldsymbol{K}_i, \boldsymbol{K}_j) = \underset{\substack{1 \leq k \leq N + (1-l)r \\ k \neq i}}{\operatorname{argmax}} s(\boldsymbol{K}_i, \boldsymbol{K}_k)) \tag{26}$$

$$= \frac{N - lr}{N + (1-l)r - 1}. \tag{27}$$

$$p(\boldsymbol{X} = \text{not}) = 1 - p(\boldsymbol{X} = \text{can}). \tag{28}$$

Denote $\boldsymbol{L} \sim p(l)$ as a discrete random variable ($\boldsymbol{L} \perp\!\!\!\perp \boldsymbol{X}$) about the current layer number, and

$$\forall 1 \leq l \leq L, \quad p(\boldsymbol{L} = l) = \frac{1}{L} \tag{29}$$

Denote $h(\boldsymbol{X}, \boldsymbol{L})$ as a indicator function

$$h(\boldsymbol{X}, \boldsymbol{L}) = \begin{cases} 1 & \text{for } \boldsymbol{X} = \text{can} \\ 0 & \text{for } \boldsymbol{X} = \text{not} \end{cases} \tag{30}$$

Then the expectation of a token from $\boldsymbol{T}^S$ can find its optimal destination token in $\boldsymbol{T}^D$ is

$$\mathbb{E}^C = \mathbb{E}_{XL}[h(\boldsymbol{X}, \boldsymbol{L})] = \sum_{l \in \boldsymbol{L}} \sum_{x \in \boldsymbol{X}} h(x, l) p_{XL}(x, l) \tag{31}$$

$$= \sum_{l \in \boldsymbol{L}} \sum_{x \in \boldsymbol{X}} h(x, l) p_{\boldsymbol{X}}(x) p_{\boldsymbol{L}}(l) \tag{32}$$

$$= \sum_{l=1}^{L} [1 \cdot p(\boldsymbol{X} = \text{can}) + 0 \cdot p(\boldsymbol{X} = \text{not})] \frac{1}{L} \tag{33}$$

$$= \frac{1}{L} \sum_{l=1}^{L} \frac{N - lr}{N + (1-l)r - 1} \tag{34}$$

Similarly, given by *bipartite soft matching* used in ToMe (Bolya et al., 2023), the expectation of a token from $\boldsymbol{T}^S$ ($|\boldsymbol{T}^S| = \lceil \frac{N + (1-l)r}{2} \rceil$) can find its optimal destination token in $\boldsymbol{T}^D$ ($|\boldsymbol{T}^D| = \lfloor \frac{N + (1-l)r}{2} \rfloor$) is

$$\mathbb{E}^B = \frac{1}{L} \sum_{l=1}^{L} \frac{1}{N + (1-l)r - 1} \lfloor \frac{N + (1-l)r}{2} \rfloor \tag{35}$$

Compare $\mathbb{E}^C$ given by *complete-graph soft matching* with $\mathbb{E}^B$ give by *bipartite soft matching*:

$$\mathbb{E}^C - \mathbb{E}^B = \frac{1}{L} \sum_{l=1}^{L} \frac{1}{N + (1-l)r - 1} (N - lr - \lfloor \frac{N + (1-l)r}{2} \rfloor) \tag{36}$$

$$\geq \frac{1}{L} \sum_{l=1}^{L} \frac{1}{N + (1-l)r - 1} (N - lr - \frac{N + (1-l)r}{2}) \tag{37}$$

$$= \frac{1}{L} [\underbrace{\sum_{l=1}^{L-1} \frac{1}{N + (1-l)r - 1} \frac{N - lr - r}{2}}_{Part1:\ 1 \leq l \leq L-1} + \underbrace{\frac{1}{N + (1-L)r - 1} \frac{N - Lr - r}{2}}_{Part2:\ l=L}] \tag{38}$$

For part1 in Eq.38, since the number of remaining tokens is always a positive integer, we have

$$N - Lr \geq 1, \tag{39}$$

and therefore for $1 \leq l \leq L - 1$:

$$N - lr \geq r + 1 \Leftrightarrow N - lr - r \geq 1 > 0 \tag{40}$$

always holds. Morever

$$N + (1 - l)r - 1 = (N - lr - 1) + r > 0 \tag{41}$$

always holds. Furthermore, we have part1 $> 0$ always holds, which indicates the expectation given by *complete-graph soft matching* is higher than *bipartite soft matching* except for the last layer.

For part2 in Eq.38, *bipartite soft matching* evenly divides tokens into two disjoint sets, and the size of each set is not less than $r$. However, the remaining tokens before the last layer may be less than $2r$. In such a situation, *bipartite soft matching* have to reduce the $r$ to the $r^\star$ such that

$$N - Lr^\star = r^\star \tag{42}$$

*complete-graph soft matching* follows the same design, and therefore part2 $= 0$ holds. Overall, we have

$$\mathbb{E}^C - \mathbb{E}^B > 0 \tag{43}$$

always holds. For example, for a CLIP (Radford et al., 2021) model with

$$N = 197, \ L = 12, \ r = 16 \tag{44}$$

used in our experiments, given by *complete-graph soft matching*, the expectation of optimal matching probability for a token from $\boldsymbol{T}^S$ is

$$\mathbb{E}^C = \frac{1}{12} \sum_{l=1}^{12} \frac{197 - l \times 16}{197 + (1 - l) \times 16 - 1} \approx 0.78, \tag{45}$$

while given by *bipartite soft matching*, the corresponding expectation is

$$\mathbb{E}^B = \frac{1}{12} \sum_{l=1}^{12} \frac{1}{197 + (1 - l) \times 16 - 1} \lfloor \frac{197 + (1 - l) \times 16}{2} \rfloor = 0.50 < \mathbb{E}^C \tag{46}$$

### A.6 COMPLEXITY ANALYSIS FOR COMPLETE-GRAPH SOFT MATCHING

Since the *sort* and *argsort* operations in Algorithm 1 and 2 can be solved by algorithms with $\mathcal{O}(N \log N)$ complexity such as *QuickSort* (Hoare, 1962), the major complexity $\mathcal{O}(N^2)$ comes from the computation of cosine similarities between each pair of tokens.

A comparison of different matching methods is listed in Table 7, which demonstrates that as a parallelizable method, *CrossGET* can achieve relatively high expectation of optimal matching probability for a certain token from $\boldsymbol{T}^S$ with relatively low complexity.

Table 7: A comparison of different matching methods. Denote $N$ as the number of the original tokens, $r$ as the number of tokens to be reduced, and $T$ as the number of iterations for k-means.

| Method | Iterative | Parallelizable | Expectation of Optimal Matching Probability | Complexity |
|---|---|---|---|---|
| Greedy Search | Yes | ✗ | $1$ | $\mathcal{O}(rN^2)$ |
| K-Means | Yes | ✗ | $\leq 1$ | $\mathcal{O}(rNT)$ |
| Random | No | ✓ | $\frac{1}{N-1} \approx 0$ | $\mathcal{O}(r)$ |
| ToMe (Bolya et al., 2023) | No | ✓ | $\frac{1}{L} \sum_{l=1}^{L} \frac{1}{N+(1-l)r-1} \lfloor \frac{N+(1-l)r}{2} \rfloor \approx 0.5$ | $\mathcal{O}(N^2)$ |
| CrossGET (Ours) | No | ✓ | $0.5 < \frac{1}{L} \sum_{l=1}^{L} \frac{N-lr}{N+(1-l)r-1} \leq 1$ | $\mathcal{O}(N^2)$ |

## A.7 DIAGRAM OF ADDING CROSS TOKES TO DIFFERENT MODELS

Figure 5: Diagram of adding cross tokes to modality-independent models such as CLIP (Radford et al., 2021) (left) and modality-dependent models such as BLIP/BLIP2 (Li et al., 2022; 2023b) (right).

Figure 5 demonstrates that *CrossGET* is designed to be a universal framework that can be used for accelerating both modality-independent vision-language models such as CLIP (Radford et al., 2021) model and modality-dependent vision-language models such as BLIP/BLIP2 (Li et al., 2022; 2023b) -based models.

Furthermore, there exist two different types of dependencies for modality-dependent vision-language models. The first type BLIP (Li et al., 2022) belongs to is that the latter modality interacts with the final output of the former modality through Cross-Attentions. For the first type, in addition to reducing the number of its own tokens, the later modality can also be accelerated by reducing the number of tokens output from the former modality to speed up Cross-Attentions in the later modality.

The second type BLIP2 (Li et al., 2023b) -based models such as InstructBLIP (Dai et al., 2023), MiniGPT-4 (Zhu et al., 2023), and mPLUG-Owl (Ye et al., 2023) belong to is that the latter modality takes the final output of the former modality as part of its input sequence to the first layer, and the cross-modal interaction is conducted by Self-Attentions in the latter modality. For the second type, the latter modality can be accelerated by reducing the length of the cross-modal input sequence to speed up Self-Attentions as well as FFNs in the latter modality.

Take BLIP2 as an example. BLIP-2 consists of a ViT for processing visual input, a Q-Former(a Bert) for bridging modalities, and an LLM for taking inputs from Q-Former and generating text output accordingly. During fine-tuning, ViT and Q-Former are tunable while the LLM is frozen. To accelerate BLIP-2, we can reduce tokens in both ViT and Q-Former. Again, it is worth noting that there are two ways to reduce the number of tokens processed by LLM. The first one is reducing the number of tokens in Q-Former. Since LLM takes Q-Former's output as its input, the token reduction conducted in Q-Former leads to LLM acceleration. The performance we reported in the paper is given by this setting. The second one is directly ensembling tokens in LLM. We have also tested this setting and discussed it in Appendix C.14.

# B SUPPLEMENTARY RELATED WORKS

**Token Reduction**    Some prior works have made progress in unimodal scenarios, for example, token reduction for vision scenarios (Chen et al., 2021; Rao et al., 2021; Su et al., 2022; Chavan et al., 2022; Liang et al., 2022b; Yin et al., 2022; Liang et al., 2022a; Bolya et al., 2023) or language scenarios (Goyal et al., 2020; Kim & Cho, 2020; Kim et al., 2022; Lassance et al., 2021). *CrossGET* is one of the early works on token reduction framework for multimodal scenarios. It is also one of the few approaches that do not require any additional learnable parameters other than negligible cross tokens. A similar work to *CrossGET* from this point of view is the unimodal approach ToMe (Bolya et al., 2023). In addition to the proposed *cross-guided matching and ensemble* specially designed for effectively exploiting cross-modal information as guidance in multimodal scenarios, *CrossGET* also proposes an improved *complete-graph soft matching* which provides more reliable token-matching results than the *bipartite soft match* adopted by ToMe, and are verified by experimental results and theoretical analysis simultaneously.

**Parameter-Efficient Fine-Tuning**    Parameter-efficient fine-tuning (PEFT) aims to reduce the number of learnable parameters while fine-tuning. It includes adapters (Houlsby et al., 2019; Sung et al., 2022b), prompt tuning (Li & Liang, 2021; Khattak et al., 2022), low-rank adaptation (Hu et al., 2021; Hyeon-Woo et al., 2021), parameter sharing (Lan et al., 2019; Shi et al., 2021), dropout (Fan et al., 2019; Shi et al., 2022) and their combinations (He et al., 2021; Karimi Mahabadi et al., 2021). LST (Sung et al., 2022a) suggests a side tuning for memory efficiency. Although parameter-efficient fine-tuning provides high efficiency during fine-tuning, the inference of the model is not accelerated. On the other hand, *CrossGET* pursues efficiency during inference, and accordingly, the model inference can be significantly accelerated.

# C    SUPPLEMENTARY EXPERIMENTS AND DETAILS

## C.1    HYPERPARAMETER SETTINGS

Table 8: Training hyperparameters for accelerating BLIP-based models.

| Hyperparameters | BLIP-NLVR (Li et al., 2022) | BLIP-Captioning (Li et al., 2022) | | BLIP-VQA (Li et al., 2022) |
|---|---|---|---|---|
| | NLVR2 (Suhr et al., 2018) | COCO Caption (Chen et al., 2015) | NoCaps (Agrawal et al., 2019) | VQAv2 (Goyal et al., 2017) |
| Optimizer | AdamW(Loshchilov & Hutter, 2017) | | | |
| AdamW $\beta$ | (0.9, 0.999) | | | |
| Batch size | 512 | | | |
| Weight decay | 0.05 | 0.05 | 0.05 | 0.05 |
| Epochs | 15 | 5 | 5 | 10 |
| Initial learning rate | $3 \times 10^{-6}$ | $1 \times 10^{-5}$ | $1 \times 10^{-5}$ | $2 \times 10^{-5}$ |
| Learning rate schedule | CosineLRScheduler (Loshchilov & Hutter, 2016) | | | |
| Data augmentation | RandomAugment (Cubuk et al., 2020) | | | |
| Training Precision | Mixed Precision (Micikevicius et al., 2017) | | | |
| Matching loss coefficient | $10^1$ | $10^2$ | $10^2$ | $10^1$ |

Table 9: Training hyperparameters for accelerating CLIP and BLIP2-based models.

| Hyperparameters | CLIP-Retrieval (Radford et al., 2021) | BLIP2-OPT2.7B-Captioning (Li et al., 2023b) | BLIP2-OPT6.7B-Captioning (Li et al., 2023b) |
|---|---|---|---|
| | Flickr30K (Young et al., 2014) | COCO Caption (Chen et al., 2015) | COCO Caption (Chen et al., 2015) |
| Optimizer | AdamW (Loshchilov & Hutter, 2017) | | |
| AdamW $\beta$ | (0.9, 0.999) | | |
| Batch size | 512 | 1024 | 512 |
| Weight decay | 0.2 | 0.05 | 0.05 |
| Epochs | 12 | 5 | 5 |
| Initial learning rate | $1 \times 10^{-5}$ | $1 \times 10^{-5}$ | $1 \times 10^{-5}$ |
| Learning rate schedule | CosineLRScheduler (Loshchilov & Hutter, 2016) | | |
| Data augmentation | RandomAugment (Cubuk et al., 2020) | | |
| Training Precision | Mixed Precision (Micikevicius et al., 2017) | | |
| Matching loss coefficient | $10^0$ | $10^{-1}$ | $10^{-1}$ |

Table 10: Structure hyperparameters for all models used in our experiments. The superscript $^*$ indicates 2 Transformers share parameters. The superscript $^\dagger$ indicates hyperparameters are from (OPT, Q-Former).

| Model | Input resolution | Vision Transformer (Touvron et al., 2021; Fang et al., 2023) | | | | Language Transformer (Devlin et al., 2018; Zhang et al., 2022) | | | |
|---|---|---|---|---|---|---|---|---|---|
| | | number | layers | width | heads | number | layers | width | heads |
| CLIP | 336×336 | 1 | 12 | 768 | 12 | 1 | 12 | 512 | 8 |
| BLIP-NLVR | 384×384 | $2^*$ | (12, 12) | (768, 768) | (12, 12) | 1 | 12 | 768 | 12 |
| BLIP-Captioning | 384×384 | 1 | 12 | 768 | 12 | 1 | 12 | 768 | 12 |
| BLIP-NoCaps | 384×384 | 1 | 12 | 768 | 12 | 1 | 12 | 768 | 12 |
| BLIP-VQA | 480×480 | 1 | 12 | 768 | 12 | 2 | (12, 12) | (768, 768) | (12, 12) |
| BLIP2-OPT2.7B | 364×364 | 1 | 39 | 1408 | 16 | $2^\dagger$ | (32, 12) | (2560, 768) | (32, 12) |
| BLIP2-OPT6.7B | 364×364 | 1 | 39 | 1408 | 16 | $2^\dagger$ | (32, 12) | (4096, 768) | (32, 12) |

The hyperparameters about model training are listed in Table 8 and Table 9. The hyperparameters about model structures are listed in Table 10.

## C.2 ABLATION STUDY ON TRAINING HYPERPARAMETERS

The hyperparameters are basically inherited from original models and do not need a specific tune. The particular case is that batch sizes are adjusted to fit our computational resources. The only additional hyperparameter introduced by *CrossGET* is the matching loss coefficient $10^\alpha$, which is used to balance the original loss items and the matching loss item. The $\alpha$ can be directly determined as the integer that makes the original loss items and the matching loss item have the closest order of magnitude, and therefore, it does not need to be tuned either.

Table 11: Ablation study about batch size on BLIP-NLVR.

| Batch size | Dev Acc | Test Acc |
|---|---|---|
| 128 | $82.0_{\downarrow0.1}$ | $82.8_{\downarrow0.4}$ |
| 256 | $82.2_{\uparrow0.1}$ | $83.0_{\downarrow0.2}$ |
| 512 | $82.1$ | $\mathbf{83.2}$ |
| 1024 | $82.2_{\uparrow0.1}$ | $83.0_{\downarrow0.2}$ |

Table 12: Ablation study about learning rate on BLIP-NLVR.

| Learning rate | Dev Acc | Test Acc |
|---|---|---|
| $1 \times 10^{-6}$ | $81.8_{\downarrow0.3}$ | $82.5_{\downarrow0.7}$ |
| $3 \times 10^{-6}$ | $82.1$ | $\mathbf{83.2}$ |
| $1 \times 10^{-5}$ | $82.2_{\uparrow0.1}$ | $82.7_{\downarrow0.5}$ |
| $3 \times 10^{-5}$ | $82.0_{\downarrow0.1}$ | $82.6_{\downarrow0.6}$ |

Table 13: Ablation study about coefficient $10^\alpha$ for matching loss on BLIP-NLVR.

| Coefficient | Dev Acc | Test Acc |
|---|---|---|
| $10^0$ | $82.0_{\downarrow0.1}$ | $82.5_{\downarrow0.7}$ |
| $10^1$ | $82.1$ | $\mathbf{83.2}$ |
| $10^2$ | $81.8_{\downarrow0.3}$ | $82.7_{\downarrow0.5}$ |

Table 11, Table 12, and Table 13 investigate how hyperparameters affect the model performance. Experimental results show that the performance is insensitive to batch size and slightly sensitive to the learning rate. As for the matching loss coefficient $10^\alpha, \alpha \in \mathbb{N}$, set it to the value that makes the original loss items and the matching loss item have the closest order of magnitude as mentioned above will work well.

## C.3 ABLATION STUDY ON DIFFERENT MODALITIES

Table 14: Ablation study about applying *CrossGET* on different modalities.

| Modality | I2T R@1 | T2I R@1 | GFLOPs |
|---|---|---|---|
| vision only | 92.1 | 79.6 | 12.0 |
| language only | $\mathbf{92.8}_{\uparrow0.7}$ | $\mathbf{80.4}_{\uparrow0.8}$ | $19.3_{\uparrow61\%}$ |
| vision and language | $91.4_{\downarrow0.7}$ | $78.3_{\downarrow1.3}$ | $\mathbf{10.6}_{\downarrow12\%}$ |

As shown in Figure 2, it is flexible that *CrossGET* can be applied on both vision and language modalities or only on one of the modalities. Table 14 investigates the trade-off between model performance and computational cost of application on different modalities. Experimental results show that *CrossGET* only on the vision modality achieves the best trade-off.

## C.4 ABLATION STUDY ON THE STRATEGY OF ADDING CROSS TOKEN

Table 15: Ablation study about the strategy of adding cross tokens.

| Depth | I2T R@1 | T2I R@1 | GFLOPs |
|---|---|---|---|
| shallow | $91.5_{\downarrow0.6}$ | $79.5_{\downarrow0.1}$ | 12.0 |
| deep | $\mathbf{92.1}$ | $\mathbf{79.6}$ | 12.0 |
| share | $90.7_{\downarrow1.4}$ | $78.8_{\downarrow0.8}$ | 12.0 |

There are several strategies for injecting cross tokens into the model. For example, (1) deep: adding different cross tokens for each layer; (2) shallow: only adding one cross token into the first layer; (3) share: adding one cross token but jointly optimized in each layer. Table 15 shows that adding different cross tokens for each layer achieves the best performance.

## C.5 ABLATION STUDY ON THE INITIALIZATION OF CROSS TOKEN

For fine-tuning, cross tokens are kind of sensitive to the initialization strategy. Using informative tokens to initialize cross tokens is recommended. More specifically, for the vision modality, [CLS]

Table 16: Ablation study about initializing cross tokens.

| Initialization | I2T R@1 | T2I R@1 | GFLOPs |
|---|---|---|---|
| zero | $91.7_{\downarrow0.4}$ | $77.9_{\downarrow1.7}$ | 12.0 |
| normal random | $90.4_{\downarrow1.7}$ | $77.6_{\downarrow2.0}$ | 12.0 |
| uniform random | $90.2_{\downarrow1.9}$ | $77.6_{\downarrow2.0}$ | 12.0 |
| informative tokens | **92.1** | **79.6** | 12.0 |

token can be used to initialize the cross token. For the language modality, the cross token can be initialized by [CLS]/[EOS]/[EOT] tokens for discriminative tasks (it depends on which token is ultimately used to calculate the loss) and by [BOS] token for auto-regressive tasks (if there is no, we use the first token of the input sequence to initialize instead).

Table 16 shows that zero initialization and random initialization perform worse. We think the sensitivity should be attributed to the limited training time for fine-tuning and the purpose of quickly adapting to downstream tasks. More specifically, random/zero initialization may work well for pre-training since there is enough time for cross tokens to learn informative guidance. However, it will be difficult for random/zero initialization to learn well with limited iterations for fine-tuning. Therefore, initializing the cross token with [CLS] token in the vision modality and [CLS]/[BOS]/[EOS]/[EOT] token in the language modality implies that the cross token already contains some informative guidance of the modality it is in, and would be easier to form more informative guidance with this good starting point.

## C.6 ABLATION STUDY ON THE PROJECTION LAYER DETACH

Table 17: Ablation study about projection layer detach.

| Projection detach | I2T R@1 | T2I R@1 | GFLOPs |
|---|---|---|---|
| neither | $91.5_{\downarrow0.6}$ | $78.9_{\downarrow0.7}$ | 12.0 |
| vision only | $91.4_{\downarrow0.7}$ | $79.4_{\downarrow0.2}$ | 12.0 |
| language only | $90.5_{\downarrow1.6}$ | $78.9_{\downarrow0.7}$ | 12.0 |
| both | **92.1** | **79.6** | 12.0 |

The final projection layers are initially used for projecting features from the different modalities into aligned representations. In *CrossGET*, the final projection layers are detached from the original model and used for aligning cross tokens. The detach operation prevents gradients with respect to cross tokens from updating the projection layers. Table 16 shows that both detaching vision and language projection give improved performance.

## C.7 ABLATION STUDY ON THE NUMBER OF CROSS TOKENS

Table 18: Ablation study on number of cross tokens.

| Number | Dev Acc | Test Acc | GFLOPs |
|---|---|---|---|
| 1 | 82.1 | **83.2** | **61.1** |
| 2 | $82.2_{\uparrow0.1}$ | $83.2_{\uparrow0.0}$ | $61.4_{\uparrow0.3}$ |
| 3 | $81.9_{\downarrow0.2}$ | $83.2_{\downarrow0.0}$ | $61.8_{\uparrow0.7}$ |
| 4 | $82.0_{\downarrow0.1}$ | $82.9_{\downarrow0.3}$ | $62.2_{\uparrow1.1}$ |

Table 18 investigates how the performance is impacted by the number of cross tokens on the Vision Reasoning Task and BLIP (Li et al., 2022) model. It can be observed that the performance is not sensitive to the increase in the number of tokens, which is unlike prompt tuning (Lester et al., 2021; Jia et al., 2022) that model performance can be boosted by increasing the number of tokens. Considering the additional computational cost of multiple cross tokens, using only one cross token is recommended.

## C.8 ABLATION STUDY ON TOKENS FOR COMPUTING IMPORTANCE

For BLIP (Li et al., 2022) on the Visual Reasoning task, a different setting from default is that not cross tokens alone, but all tokens are used to compute importance. By default, in the modality-independent model CLIP (Radford et al., 2021), only the [CLS] and [EOS] tokens are ultimately used for computing loss. In contrast, for the modality-dependent model BLIP-NLVR, all tokens output from the vision modality are parts of the inputs for the language modality and matter.

Table 19: Ablation study about tokens for computing importance.

| used tokens | Dev Acc | Test Acc |
|---|---|---|
| cross token | $82.1_{\uparrow 0.0}$ | $82.9_{\downarrow 0.3}$ |
| other tokens | $81.9_{\downarrow 0.2}$ | $82.2_{\downarrow 1.0}$ |
| all tokens | $82.1$ | **83.2** |
| importance | $82.2_{\uparrow 0.1}$ | $83.0_{\downarrow 0.2}$ |

Four settings about which tokens are used for computing importance are tested as shown in Table 19: (1) cross token: cross tokens contribute all; (2) other tokens: other tokens contribute all; (3) all tokens (adopted): cross tokens contribute to $\frac{1}{2}$ importance while other tokens contribute to the other $\frac{1}{2}$; (4) importance: we can reuse the dot product between the query and key of each token (including cross tokens) that has already been calculated in the Self-Attention as the importance metric to avoid extra computational cost for introducing other tokens' importance.

## C.9 ABLATION STUDY ON TOKENS FOR COMPUTING JS DIVERGENCE

Table 20: Ablation study about which tokens are used for computing JS divergence as additional loss items.

| JS divergence as loss | CIDEr | SPICE |
|---|---|---|
| only between pairs of cross tokens | $130.2_{\downarrow 1.4}$ | $23.7_{\downarrow 0.1}$ |
| only between cross tokens and other tokens | $131.0_{\downarrow 0.6}$ | $23.5_{\downarrow 0.3}$ |
| w/o weighting loss according to generation order | $131.2_{\downarrow 0.4}$ | $23.7_{\downarrow 0.1}$ |
| between cross tokens and all tokens | **131.6** | **23.8** |

For BLIP (Li et al., 2022) on the Image Caption task, a different setting from default is that not only the loss of JS divergence between the pairs of cross tokens but also the JS divergence between the cross tokens and other tokens should be added as loss items. By default, in the language modality of the discriminative model CLIP (Radford et al., 2021), only the [EOS] token matters for the final output. In contrast, for the auto-regressive model BLIP-Captioning, tokens are generated based on their previous tokens, and therefore, every token matters.

Table 20 shows that combined JS divergence between pairs of cross tokens as well as between cross tokens and other tokens as loss performs best. Besides, weighting the loss between cross tokens and other tokens according to the generation order also helps. The weight for the $i$-th generated token is $1 - \frac{i}{L}$ where $L$ is the maximum generation length, which means the first generated token is more important than the later ones since they are generated based on former ones.

## C.10 COMPARISON EXPERIMENTS WITH TEXT-RELEVANT IMAGE PATCH SELECTION

In Table 21, the TRIPS (Default FLOPs) (Jiang et al., 2022) indicates we follow the recommended setting of the original TRIPS, *i.e.*, we take the 5th and 10th as the patch-selection layer and set the keep ratio of each layer to 70%. The TRIPS (Same FLOPs) indicates we decrease the keep ratio of each patch-selection layer to achieve similar GFLOPs with ToMe (Bolya et al., 2023) and CrossGET. Overall, the experimental results demonstrate that **CrossGET outperforms TRIPS under similar computational costs**.

When compared with TRIPS, one of the differences is that **CrossGET can more easily deal with the models in which the embedding sizes of vision and language branches are different**, which is also an important contribution of CrossGET. More specifically, TRIPS is not directly suitable for

Table 21: Accelerate CLIP on the Flickr30K dataset of the Image-Text Retrieval task. R: Recall. R@1, R@5, and R@10 are the higher the better. The TRIPS represents Text-Relevant Image Patch Selection (Jiang et al., 2022). The -L indicates using an additional learnable projection to align the text [CLS] token with vision tokens.

| Approach | Image → Text | | | Text → Image | | | Avg. | GFLOPs | Throughput |
|---|---|---|---|---|---|---|---|---|---|
| | R@1 | R@5 | R@10 | R@1 | R@5 | R@10 | $\overline{\text{R@1}}$ | ↓ | ↑ |
| CLIP (Radford et al., 2021) | 92.1 | 99.1 | 99.7 | 79.3 | 95.7 | 98.0 | 85.7 | 20.6 | 255.2 |
| TRIPS (Default FLOPs) | 87.6 | 98.7 | 99.4 | 76.6 | 94.4 | 97.0 | 82.1 | 16.4 | 317.7 |
| TRIPS-L (Default FLOPs) | 90.4 | 98.9 | 99.5 | 76.8 | 94.4 | 97.2 | 83.6 | 16.4 | 316.9 |
| TRIPS (Same FLOPs) | 75.5 | 94.3 | 97.8 | 63.9 | 88.5 | 93.8 | 69.7 | 12.0 | 423.5 |
| TRIPS-L (Same FLOPs) | 70.1 | 92.4 | 96.9 | 61.2 | 86.8 | 92.1 | 65.7 | 12.0 | 423.1 |
| CrossGET (Ours) | $\mathbf{92.1}_{\uparrow 0.0}$ | $99.7_{\uparrow 0.6}$ | $99.8_{\uparrow 0.1}$ | $\mathbf{79.6}_{\uparrow 0.3}$ | $95.7_{\uparrow 0.0}$ | $98.0_{\uparrow 0.0}$ | $\mathbf{85.9}_{\uparrow 0.2}$ | $12.0_{\downarrow 42\%}$ | $401.8_{\uparrow 57\%}$ |

the models with different embedding sizes of the vision branch and language branch, and without a projection layer that projects the language embedding size into the vision embedding size as well.

For example, the vision and language embedding size in the CLIP model we used is 768 and 512, respectively. Besides, there is a $768 \rightarrow 512$ projection layer for vision projection and a $512 \rightarrow 512$ for language projection. TRIPS requires the projected text [CLS] token to have the same embedding size as the tokens in the vision branch. However, there is no trained (*i.e.*, aligned) $512 \rightarrow 768$ projection layer in CLIP to fulfill this. To overcome this problem, we propose two strategies: (1) The first one is to use the pseudo inverse of the trained $768 \rightarrow 512$ projection layer to project the 512-dimensional text [CLS] token into a 768-dimensional token, whose experimental results are denoted without -L. (2) The second one is to add an additional $512 \rightarrow 768$ learnable projection layer into the original model and then jointly optimize, whose experimental results are denoted with -L. On the contrary, **this is not a problem for CrossGET** since cross tokens are learned cross-modally while used intra-modally, and the embedding size of cross tokens is the same as other tokens in the same modality branch. Thus, CrossGET doesn't need a projection layer to align cross tokens when they are used as metrics to guide the token reduction.

When compared with TRIPS, the other additional difference is that **CrossGET can easily deal with both modality-independent and modality-dependent models**, which is also an important contribution of CrossGET. We have discussed this in the "Exploiting Cross-Modal Guidance" paragraph of Section 3.2, and TRIPS can serve as an example to elaborate it. More specifically, TRIPS uses text [CLS] token, *i.e.*, the output of the language encoder in the ALBEF (Li et al., 2021) model as the metric to guide the token reduction in the vision branch. However, this paradigm cannot be used in multimodal models in that the input of the language branches depends on the output of the vision branch.

For example, in the BLIP-NLVR (Li et al., 2022) model, the output of the vision branch is a necessary input for the language branch. And if we want to use the text [CLS] token *i.e.*, the output of the language branch as a metric to guide the token reduction in the vision branch, we have to first forward through the vision branch, get the last layer's output as the input of the language branch, forward through the language branch, get the last layer's output as the metric, *i.e.*, only after the forward of the vision branch is finished, we can get the required metric used for vision branch. **CrossGET breaks this paradox of cycles** by using cross tokens as agents for other modalities, providing cross-modal guidance on behalf of other modalities without inference on other modalities.

## C.11 COMPARISON EXPERIMENTS WITH ADPATER

The experimental results demonstrate that when using ToMe (Bolya et al., 2023) with an adapter (Chen et al., 2022), the middle dimension of the adapter needs to be very large(*e.g.*, around 4096) for the model to perform better than without using the adapter. However, the additional computation cost introduced by the adapter is significant (see GFLOPs and Throughput in the above Table 22), and the performance is still worse than CrossGET.

Table 22: Accelerate CLIP on the Flickr30K dataset of the Image-Text Retrieval task. R: Recall. R@1, R@5, and R@10 are the higher the better. The Adapter-x represents Adaptformer (Chen et al., 2022), and the integer x in Adapter-x represents the middle dimension of the adapter.

| Approach | Image → Text | | | Text → Image | | | Avg. R@1 | GFLOPs ↓ | Throughput ↑ |
|---|---|---|---|---|---|---|---|---|---|
| | R@1 | R@5 | R@10 | R@1 | R@5 | R@10 | | | |
| CLIP (Radford et al., 2021) | 92.1 | 99.1 | 99.7 | 79.3 | 95.7 | 98.0 | 85.7 | 20.6 | 255.2 |
| ToMe (Bolya et al., 2023) | 90.8 | 99.2 | 99.5 | 78.1 | 95.3 | 97.7 | 84.5 | 11.8 | 417.4 |
| ToMe❄+Adapter-16🔥 | 89.2 | 98.7 | 99.6 | 75.9 | 94.2 | 97.1 | 82.6 | 11.9 | 404.1 |
| ToMe❄+Adapter-64🔥 | 89.9 | 98.8 | 99.5 | 76.7 | 94.4 | 97.3 | 83.3 | 12.0 | 401.2 |
| ToMe❄+Adapter-256🔥 | 90.2 | 99.0 | 99.4 | 76.7 | 94.5 | 97.5 | 83.5 | 12.5 | 386.4 |
| ToMe❄+Adapter-1024🔥 | 90.3 | 99.0 | 99.8 | 78.0 | 94.6 | 97.4 | 84.2 | 14.5 | 346.2 |
| ToMe❄+Adapter-4096🔥 | 91.4 | 98.8 | 99.6 | 78.2 | 95.0 | 97.6 | 84.8 | 22.7 | 243.5 |
| CrossGET (Ours) | $92.1_{\uparrow0.0}$ | $99.7_{\uparrow0.6}$ | $99.8_{\uparrow0.1}$ | $79.6_{\uparrow0.3}$ | $95.7_{\uparrow0.0}$ | $98.0_{\uparrow0.0}$ | $85.9_{\uparrow0.2}$ | $12.0_{\downarrow42\%}$ | $401.8_{\uparrow57\%}$ |

## C.12 EXPERIMENTS WITH COOP ON THE FEW-SHOT IMAGE CLASSIFICATION TASK

We conduct experiments on ImageNet (Deng et al., 2009), Caltech101 (Fei-Fei et al., 2004), and OxfordPets (Parkhi et al., 2012) datasets. We follow the same setting from CoOp (Zhou et al., 2022) that uses 16 shots and freezes the backbone model CLIP while conducting prompt tuning.

Table 23: Accelerate CoOp on the ImageNet dataset of the few-shot Image Classification task.

| Method | Top-1 Acc (%) | GFLOPs | Method | Top-1 Acc (%) | GFLOPs |
|---|---|---|---|---|---|
| CoOp (16 shots) | 71.1 | 20.6 | CoOp (16 shots) | 71.1 | 20.6 |
| ToMe | 70.4 | 16.2 | CrossGET | **71.0** | 16.5 |
| ToMe | 68.9 | 14.0 | CrossGET | **70.2** | 14.2 |
| ToMe | 62.8 | 11.8 | CrossGET | **67.4** | 12.0 |

Table 24: Accelerate CoOp on the Caltech101 dataset of the few-shot Image Classification task.

| Method | Top-1 Acc (%) | GFLOPs | Method | Top-1 Acc (%) | GFLOPs |
|---|---|---|---|---|---|
| CoOp (16 shots) | 95.4 | 20.6 | CoOp (16 shots) | 95.4 | 20.6 |
| ToMe | 93.7 | 14.0 | CrossGET | **94.9** | 14.2 |
| ToMe | 93.1 | 11.8 | CrossGET | **94.0** | 12.0 |

Table 25: Accelerate CoOp on the OxfordPets dataset of the few-shot Image Classification task.

| Method | Top-1 Acc (%) | GFLOPs | Method | Top-1 Acc (%) | GFLOPs |
|---|---|---|---|---|---|
| CoOp (16 shots) | 93.3 | 20.6 | CoOp (16 shots) | 93.3 | 20.6 |
| ToMe | 89.5 | 14.0 | CrossGET | **91.1** | 14.2 |
| ToMe | 89.5 | 11.8 | CrossGET | **89.6** | 12.0 |

The experimental results in Table 23, 24 and 25 demonstrate that CrossGET also consistently outperforms ToMe on the CoOp benchmark. Besides, the performance-cost trade-off on the CoOp benchmark is relatively worse than other experiments we have reported, which should be attributed to: 1) Most of the model parameters (*i.e.*, the whole backbone) are frozen, resulting in a worse convergence status than the full-parameter fine-tuning we have used for other experiments. 2) Only a part of the datasets are used for few-shot learning, resulting in severer overfitting than the entire datasets we have used for other experiments.

## C.13 EXPERIMENTS WITH BLIP2 ON THE IMAGE CAPTIONING TASK

Experimental results on BLIP2-OPT2.7B (Li et al., 2023b) are listed in Table 26, which demonstrates similarly promising performance of *CrossGET* as on BLIP2-OPT6.7B. Note that the performance

of the original model that we tested by ourselves is slightly lower than the results reported in the original paper.

Table 26: Accelerate multimodal LLM BLIP2-OPT2.7B (Li et al., 2023b) on the COCO Caption dataset of the Image Caption task. The suffix -F denotes GFLOPs and throughput for the forward, while -G denotes GFLOPs and throughput for the generation.

| Approach | Tuning | CIDEr | BLEU@4 | GFLOPs-F | Tput.-F | GFLOPs-G | Tput.-G |
|---|---|---|---|---|---|---|---|
| BLIP2-OPT2.7B | - | 145.6 | 42.8 | 854.2 | 54.0 | 1379.3 | 22.3 |
| ToMe (Bolya et al., 2023) | w/o tuning | 145.1 | 42.6 | 769.1 | - | 1294.2 | - |
| | w/o tuning | 144.2 | 42.3 | 679.7 | - | 1218.1 | - |
| | w/o tuning | 142.8 | 42.2 | 592.2 | - | 1104.0 | - |
| | w/o tuning | 136.5 | 40.6 | 506.7 | - | 1018.5 | - |
| | w/ tuning | $142.4_{\downarrow 3.2}$ | $41.7_{\downarrow 1.1}$ | 404.6 | 107.5 | 855.1 | 30.5 |
| CrossGET(Ours) | w/o tuning | 145.9 | **43.1** | 785.4 | - | 1310.5 | - |
| | w/o tuning | 144.6 | **42.6** | 692.7 | - | 1204.5 | - |
| | w/o tuning | 144.2 | **42.7** | 602.5 | - | 1114.3 | - |
| | w/o tuning | 138.6 | **41.2** | 514.9 | - | 1053.3 | - |
| | w/ tuning | $\mathbf{143.1}_{\downarrow 2.5}$ | $\mathbf{41.9}_{\downarrow 0.9}$ | $413.9_{\downarrow 52\%}$ | $104.5_{\uparrow 94\%}$ | $822.0_{\downarrow 40\%}$ | $30.5_{\uparrow 37\%}$ |

## C.14 EXPERIMENTS WITH BLIP2 ABOUT WHERE TO REDUCE TOKENS

We conduct experiments on directly ensembling tokens on OPT. To elaborate, directly ensembling tokens on OPT leads to a smaller KV cache. Therefore, during each generation step, a smaller number of previous tokens' KV cache will attend to the current token and thus need less computational cost for Self-Attentions, which is achieved by ensembling previous tokens by Complete-Graph Soft Matching and cross-modal guidance from cross tokens.

Table 27: Accelerate multimodal LLM BLIP2-OPT on the COCO Caption dataset of the Image Caption task.

| Method | Where to reduce tokens | CIDEr | GFLOPs |
|---|---|---|---|
| BLIP2-OPT2.7B | / | 145.6 | 854.2 |
| BLIP2-OPT2.7B with CrossGET | On ViT and Q-Former | 143.1 | 413.9 |
| BLIP2-OPT2.7B with CrossGET | On ViT and LLM | $142.4_{\downarrow 0.7}$ | 417.7 |
| BLIP2-OPT6.7B | / | 144.5 | 1042.6 |
| BLIP2-OPT6.7B with CrossGET | On ViT and Q-Former | 143.1 | 558.2 |
| BLIP2-OPT6.7B with CrossGET | On ViT and LLM | $143.5_{\uparrow 0.4}$ | 566.1 |

An intriguing finding from Table 27 is that the inference capabilities of OPT are positively affected by directly ensembling tokens on OPT on the relatively larger OPT6.7B model while negatively affected by the same setting on the relatively smaller OPT2.7B model. A possible explanation for the contrasting behaviors is:

- To accelerate OPT, the smaller 2.7B model is more vulnerable to the disturbance brought by the token ensemble within the model (note that the OPT model is frozen, so it cannot adapt its weights of parameters when the number of tokens is getting smaller). Therefore, applying CrossGET on Q-Former is a better setting so that the OPT model is accelerated by taking fewer input tokens.

- The larger 6.7B model is more resilient to the disturbance brought by the token ensemble within the model. Moreover, ensembling tokens within the OPT model can help preserve the tokens' information as much as possible (if the number of tokens is reduced in the preceding Q-Former, the lost information due to the ensemble operation will be inaccessible to the succeeding OPT model).

Moreover, the experiments also indicate that after ensembling tokens, at least two competing factors determine the extent of the performance affected: 1) performance increases due to preserving more

tokens' information within the OPT model. 2) performance decreases depending on frozen OPT's resilience to the disturbance brought by token ensemble within the model.

## C.15 EVALUATION AT DIFFERENT REDUCTION RATIOS WITHOUT TRAINING

Exhaustive experimental results at different reduction ratios *without* training are listed in Table 28.

Table 28: Experimental results at different reduction ratios for BLIP on the NVLR2 dataset of the Visual Reasoning task *without* training.

| Approach | Test Acc | Drop | GFLOPs | Reduction |
|---|---|---|---|---|
| BLIP (Li et al., 2022) | 83.38 | - | 132.54 | - |
| | 83.34 | -0.04 | 136.92 | 0.97x |
| | 83.32 | -0.06 | 135.18 | 0.98x |
| | 83.40 | +0.02 | 133.43 | 0.99x |
| | 83.22 | -0.16 | 131.69 | 1.01x |
| | 83.17 | -0.21 | 129.96 | 1.02x |
| | 83.02 | -0.36 | 128.23 | 1.03x |
| | 83.08 | -0.30 | 126.50 | 1.05x |
| | 82.99 | -0.39 | 124.78 | 1.06x |
| | 82.88 | -0.50 | 123.07 | 1.08x |
| | 82.81 | -0.57 | 121.36 | 1.09x |
| | 82.85 | -0.53 | 119.65 | 1.11x |
| | 82.66 | -0.72 | 117.95 | 1.12x |
| | 82.48 | -0.90 | 116.25 | 1.14x |
| | 82.20 | -1.18 | 114.56 | 1.16x |
| | 82.30 | -1.08 | 112.87 | 1.17x |
| | 82.02 | -1.36 | 111.19 | 1.19x |
| | 81.67 | -1.71 | 109.51 | 1.21x |
| | 81.90 | -1.48 | 107.84 | 1.23x |
| | 81.75 | -1.63 | 106.17 | 1.25x |
| | 81.63 | -1.75 | 104.51 | 1.27x |
| | 81.47 | -1.91 | 102.84 | 1.29x |
| | 81.43 | -1.95 | 101.19 | 1.31x |
| | 81.29 | -2.09 | 99.54 | 1.33x |
| | 80.93 | -2.45 | 97.89 | 1.35x |
| CrossGET | 80.87 | -2.51 | 96.25 | 1.38x |
| (*without* training) | 80.93 | -2.45 | 94.61 | 1.40x |
| | 80.86 | -2.52 | 92.98 | 1.43x |
| | 80.68 | -2.70 | 91.35 | 1.45x |
| | 80.55 | -2.83 | 89.73 | 1.48x |
| | 80.28 | -3.10 | 88.12 | 1.50x |
| | 80.35 | -3.03 | 86.50 | 1.53x |
| | 80.22 | -3.16 | 84.89 | 1.56x |
| | 80.22 | -3.16 | 83.29 | 1.59x |
| | 80.38 | -3.00 | 81.69 | 1.62x |
| | 80.17 | -3.21 | 80.10 | 1.65x |
| | 80.17 | -3.21 | 78.51 | 1.69x |
| | 80.30 | -3.08 | 76.92 | 1.72x |
| | 80.12 | -3.26 | 75.34 | 1.76x |
| | 80.21 | -3.17 | 73.76 | 1.80x |
| | 80.02 | -3.36 | 72.19 | 1.84x |
| | 79.79 | -3.59 | 70.63 | 1.88x |
| | 79.64 | -3.74 | 69.07 | 1.92x |
| | 80.02 | -3.36 | 67.51 | 1.96x |
| | 79.87 | -3.51 | 65.96 | 2.01x |
| | 79.68 | -3.70 | 64.60 | 2.05x |
| | 79.32 | -4.06 | 63.33 | 2.09x |
| | 79.10 | -4.28 | 62.03 | 2.14x |
| | 78.80 | -4.58 | 60.78 | 2.18x |
| | 78.85 | -4.53 | 59.73 | 2.22x |
| | 78.85 | -4.53 | 58.65 | 2.26x |

## C.16 EVALUATION AT DIFFERENT REDUCTION RATIOS WITH TRAINING

More experimental results at different reduction ratios with training are listed in Table 29.

Table 29: Experimental results at different reduction ratios for BLIP on the NVLR2 dataset of the Visual Reasoning task with training.

| Approach | Test Acc | | Drop | GFLOPs | | Reduction |
|---|---|---|---|---|---|---|
| BLIP (Li et al., 2022) | 83.38 | | - | 132.54 | | - |
| CrossGET (with training) | 83.74 | | +0.36 | 118.34 | | 1.12x |
| | 83.31 | | -0.07 | 85.27 | | 1.55x |
| | 83.19 | | -0.19 | 61.09 | | 2.17x |
| | 82.28 | | -1.10 | 58.95 | | 2.25x |
| | 81.33 | | -2.05 | 53.25 | | 2.49x |
| | 80.67 | | -2.71 | 50.14 | | 2.64x |
| | 78.19 | | -5.19 | 45.11 | | 2.94x |

## C.17 RE-EVALUATION TRAINED MODEL AT DIFFERENT REDUCTION RATIOS

Once *CrossGET* has trained a model at a certain compression ratio, a series of models with different performance and computational costs are obtained simultaneously. More specifically, by simply adjusting the number of tokens reduced at inference, it is free to use different models without training based on the desired budget. Table 30 provides the relevant experimental results for CLIP model on the Flickr30K dataset of the Image-Text Retrieval task.

Table 30: Experimental results for re-evaluating a model trained by CrossGET (50% tokens reduced) at different reduction ratios *without* training.

| Approach | Recall@1 - Trained | | Change | GFLOPs | | Increase |
|---|---|---|---|---|---|---|
| CLIP (Radford et al., 2021) | 85.70 | | -0.15 | 20.57 | | 1.71x |
| CrossGET (re-evaluate *without* training) | 85.86 | | +0.01 | 20.70 | | 1.72x |
| | 85.79 | | -0.06 | 20.48 | | 1.70x |
| | 85.93 | | +0.08 | 19.90 | | 1.65x |
| | 85.88 | | +0.03 | 19.32 | | 1.60x |
| | 86.06 | | +0.21 | 18.74 | | 1.56x |
| | 86.19 | | +0.34 | 18.17 | | 1.51x |
| | 86.34 | | +0.49 | 17.60 | | 1.46x |
| | 86.15 | | +0.30 | 17.03 | | 1.41x |
| | 86.21 | | +0.36 | 16.46 | | 1.37x |
| | 86.33 | | +0.48 | 15.90 | | 1.32x |
| | 86.19 | | +0.34 | 15.33 | | 1.27x |
| | 86.29 | | +0.44 | 14.77 | | 1.23x |
| | 86.06 | | +0.21 | 14.22 | | 1.18x |
| | 86.01 | | +0.16 | 13.66 | | 1.13x |
| | 86.08 | | +0.23 | 13.11 | | 1.09x |
| | 86.20 | | +0.35 | 12.56 | | 1.04x |
| CrossGET (with training) | 85.85 | | - | 12.04 | | - |

