# OpenReview forum: "CrossGET: Cross-Guided Ensemble of Tokens for Accelerating Vision-Language Transformers"
_ICLR.cc/2024/Conference — ICLR 2024 Conference Withdrawn Submission_

### Official Review · Reviewer_Uwik · 2023-10-31

**Soundness:** 3 good
**Presentation:** 3 good
**Contribution:** 2 fair
**Rating:** 5
**Confidence:** 5

**Summary:**

The paper introduces the Cross-Guided Ensemble of Tokens (CrossGET), which is designed to enhance the efficiency of vision-language Transformers. It tackles the significant challenge of mitigating the computational costs and latency associated with vision-language models. Within this framework, two essential components come into play: Cross-Guided Matching and Ensemble, orchestrating the fusion of tokens guided by cross-modal cues, and Complete-Graph Soft Matching, contributing to the refinement of token matching outcomes.

**Strengths:**

1.Comprehensive Experimentation and Solid Theoretical Foundation: The paper's strength lies in its extensive and well-documented experiments, combined with a rigorous theoretical underpinning for the proposed method. This makes the work sound and reliable, both in terms of its theoretical framework and practical applicability.
2. Relevance of the Addressed Problem: The choice of the problem addressed in the paper holds significant value, especially in the context of the substantial computational overhead associated with many state-of-the-art multimodal models. This highlights the practical importance of the research. However, it is recommended that the authors extend their analysis and experimentation to encompass a broader range of models, moving beyond the initial exploration with BLIP-2. This would further enhance the paper's contribution and generalizability.

**Weaknesses:**

1. Cross-Modal Guidance Utilization: In the paper, the emphasis is placed on the ability of CrossGET to be applied to modality-dependent models like BLIP and BLIP2. The approach involves learning a cross-token to serve as guidance for another modality. However, there are concerns about this approach. Taking BLIP as an example, it appears that it may not fully harness textual guidance. In scenarios like visual grounding, where different textual descriptions highlight various aspects of the same image, it raises questions about how CrossGET selects tokens from different texts to focus on.
2. Unfair Experimental Comparisons: The paper contains instances of unfair comparisons in the experiments. For example, in section 4.1, the authors directly compare retrieval results of models such as TRIPS and UPOP. Yet, these models vary significantly in terms of training data and model parameter sizes, making the comparison less meaningful. To provide a clearer perspective, the paper should emphasize how much TRIPS, or similar acceleration methods, improve over the baseline, and how much the proposed method accelerates and enhances performance compared to the baseline.
3. Limited Model Performance Improvement: The paper reports only marginal improvements in model performance while introducing a relatively complex method. Moreover, the acceleration achieved by the proposed method appears similar to that of ToMe. Given the relative complexity of the proposed approach, the effectiveness of this work may be questioned, especially if the gains in performance and acceleration are not substantial.

**Questions:**

1. Implementation of Token Reduction in BLIP-2: It would be beneficial for the authors to provide more detailed information on how they specifically implemented token reduction in BLIP-2 within the context of their method. A more elaborate explanation of the process and its impact on BLIP-2's performance would enhance the clarity and completeness of the paper.
2. Impact of CrossGET on OPT in BLIP-2: A notable aspect of this work is the introduction of CrossGET into the frozen OPT component of BLIP-2 for token reduction. However, it's important to consider that OPT is a decoder-only model. The paper should address how this approach might affect the inference capabilities of OPT and whether any experiments were conducted to analyze and verify why image captioning performance appears to be minimally impacted. Further insight into this aspect of the methodology would enhance the paper's robustness and contribute to a better understanding of the results.Im glad to improve my score if my   concerns be addressed.

---

> ### Author Response · Authors · 2023-11-22
> **Response to Reviewer Uwik [Part 1/3]**
>
> We sincerely appreciate Reviewer Uwik’s thoughtful feedback. We are deeply encouraged by Reviewer Uwik’s all-around recognition of comprehensive experiments, solid theoretical foundation, and significant value in solving practical problems. Below, we provide a detailed response with required experiments to address each concern.
>
> ---
>
> *W1. Cross-Modal Guidance Utilization.*
>
> A1.
>
> - For “*Taking BLIP as an example, it appears that it may not fully harness textual guidance*”: CrossGET uses cross tokens in each modality to serve as agents to learn information from other modalities and provide cross-modal guidance. Admittedly, even though it is hard to quantitatively calculate how much information cross tokens can harness, the positive effect on reported performance ensures the presence of successfully exploited cross-modal information, which is a leap **from non-existence to existence**. To elaborate, we would like to highlight that the proposed Cross-Guided Matching & Ensemble is **the first method** to **make it possible** that **in modality-dependent models, the preceding modality can utilize (some of the) information from the succeeding modality to guide token reduction**, which is a very useful contribution considering most of recent popular VLMs are exactly modality-dependent models (*e.g.*, InstructBLIP, MiniGPT4, and LLaVA).
> - For *“In scenarios like visual grounding, where different textual descriptions highlight various aspects of the same image, it raises questions about how CrossGET selects tokens from different texts to focus on”*: The utilization of the proposed method in visual grounding is similar to it in Image-Text Retrieval tasks. More specifically, learnable cross tokens are injected into a visual grounding model's visual and language modules individually. They are driven to learn a **general/coarse** information representation (like [CLS]/[EOS] token in some visual/language understanding models) from each other's modalities by a cross-matching loss in Eq. 2. It is worth noting that **the cross tokens are input-related**, *i.e.*, different textual descriptions **will lead to different cross tokens since cross tokens will interact with normal tokens within its modality. Guided by learned cross tokens, visual tokens unrelated to texts (*e.g.*, background pixels) and text tokens unrelated to the image (*e.g.*, most of the non-substantive words) will be selected out.
>
> ---
>
> *W2. Experimental Comparisons.*
>
> A2.
>
> - For *“Yet, these models vary significantly in terms of training data and model parameter sizes, making the comparison less meaningful”*: We would like to clarify that all of the reported models are trained from the same baseline model (*i.e.*, the TRIPS is also trained on the same CLIP in Section 4.1), use the same training data, and has the same number of parameters with negligible discrepancy (less than 0.01%). In other words, **we control all possible variables except the methods themselves to achieve fair comparisons**.
> - For *“the paper should emphasize how much TRIPS, or similar acceleration methods, improve over the baseline, and how much the proposed method accelerates and enhances performance compared to the baseline.”*: Below is the required table to demonstrate the relative improvement over their individual baselines (The data of TRIPS is directly excerpted from its paper. Since there is no reported acceleration-related data for the Image-Text Retrieval task, we conduct comparisons on NLVR and VQA tasks).
>
>
>     | Model  | NLVR dev Acc | Throughput | Model | NLVR dev Acc | Throughput |
>     | --- | --- | --- | --- | --- | --- |
>     | TRIPS’s baseline model | 82.35 | 79.32 | CrossGET’s baseline model | 82.3 | 39.8 |
>     | TRIPS | 80.45 (-1.9) | 153.92(+94%) | CrossGET | 82.1 **(-0.1)** | 76.8(+93%) |
>
>     | Model  | VQA test-dev Acc | Throughput | Model | NLVR dev Acc | Throughput |
>     | --- | --- | --- | --- | --- | --- |
>     | TRIPS’s baseline model | 76.12 | 79.32 | CrossGET’s baseline model | 77.4 | 67.2 |
>     | TRIPS | 75.29 (-0.83) | 143.37(+81%) | CrossGET | 77.0 **(-0.4)** | 120.4(+79%) |
>
>     The above tables demonstrate that CrossGET still outperforms TRIPS under a similar speedup ratio by comparison with the suggested way. Two possible questions regarding the above comparison are:
>
>     - How do we choose TRIPS’s datapoint for comparison? According to the speedup ratio (given by throughput) of CrossGET, we select the closest of TRIPS’s datapoint to the speedup ratio of CrossGET for a fair comparison.
>     - Why is the performance of TRIPS reported here seemingly better than it was reported in Section 4.1 of our paper? This should mainly be attributed to a certain degree of incompatibility between it and our baseline models. We have provided a detailed explanation to elaborate in Appendix C.10.
>
> ---

---

> ### Author Response · Authors · 2023-11-22
> **Response to Reviewer Uwik [Part 2/3]**
>
> *W3. Limited Model Performance Improvement*
>
> A3.
>
> - For *“marginal improvements in model performance”*: We would like to explain why the performance improvements should be considered as **significant instead of marginal**. It is widely known that the model performance does **not improve linearly** with the growth of the computational cost. In other words, a slight improvement in accuracy is typically at the expense of notably huge computational costs. Take experiments on NLVR2 datasets in Table 2 of our paper as an example:
>     - Compared with the original model, CrossGET achieves nearly 2x speedup with only a 0.2% accuracy drop.
>     - Compared with the previous SOTA method ToMe, which has a 1.2% accuracy drop, CrossGET reduces the accuracy drop by more than $\frac{1.2-0.2}{1.2} \approx 83\\%$ while maintaining a similar acceleration speedup.
>
>     Therefore, we believe that the performance improvements should be considered as significant instead of marginal.
>
> - For *“acceleration achieved by the proposed method”*: When we are talking about model acceleration, **what matters most is the performance-cost trade-off**, *i.e.*, it is a binary relation always involved with performance and cost/speed simultaneously. Therefore, by illustrating performance improvement under the “*similar acceleration*” is a widely recognized way to show the superiority of the proposed method. Back to our method, we have exactly validated the superiority of CrossGET by trade-off plots (*e.g.*, Figure 4) and (performance, cost) datapoints scattered in lots of tables in the paper.
> - For “*substantial effectiveness of this work, especially the gains in performance and acceleration*”:
>
>     Please don't worry about that. We have saved checkpoints and training logs matched with datapoints in the paper, and we will make them publicly available.
>
>
> ---
>
> *Q1*. *Implementation of Token Reduction in BLIP-2.* *How they specifically implemented token reduction in BLIP-2.*
>
> A1. BLIP-2 consists of a ViT for processing visual input, a Q-Former(a Bert) for bridging modalities, and an LLM for taking inputs from Q-Former and generating text output accordingly. During fine-tuning, ViT and Q-Former are tunable while the LLM is frozen. To accelerate BLIP-2, we ensemble tokens in both ViT and Q-Former. As we explained in Appendix A.7, it is worth noting that there are **two ways to reduce the number of tokens processed by LLM**. The first one is reducing the number of tokens in Q-Former. Since LLM takes Q-Former’s output as its input, the **token reduction conducted in Q-Former leads to LLM acceleration**. The performance we reported in the paper is given by this setting. The second one is directly ensembling tokens in LLM. We have also tested this setting and discuss it in the answer to the next question.
>
> ---

---

> ### Author Response · Authors · 2023-11-22
> **Response to Reviewer Uwik [Part 3/3]**
>
> Q2. *Impact of CrossGET on OPT in BLIP-2. How this approach might affect the inference capabilities of OPT and whether any experiments were conducted to analyze and verify why image captioning performance appears to be minimally impacted.*
>
> A2. To investigate the impact of CrossGET on OPT, we conduct new experiments on directly ensembling tokens on OPT. To elaborate, directly ensembling tokens on OPT leads to a **smaller KV cache**. Therefore, during each generation step, **a smaller number of previous tokens’ KV cache will attend to the current token** and thus need **less computational cost for Self-Attentions**, which is achieved by ensembling (fusing) previous tokens by Complete-Graph Soft Matching and cross-modal guidance from cross tokens.
>
> | Model | Where to reduce tokens | CIDEr ($\uparrow$) | GFLOPs |
> | --- | --- | --- | --- |
> | BLIP2-OPT2.7B | / | 145.6 | 854.2 |
> | BLIP2-OPT2.7B with CrossGET | On ViT and Q-Former | 143.1 | 413.9 |
> | BLIP2-OPT2.7B with CrossGET | On ViT and LLM | 142.4 **(-0.7)** | 417.7 |
> | BLIP2-OPT6.7B | / | 144.5 | 1042.6 |
> | BLIP2-OPT6.7B with CrossGET | On ViT and Q-Former | 143.1 | 558.2 |
> | BLIP2-OPT6.7B with CrossGET | On ViT and LLM | 143.5 **(+0.4)** | 566.1 |
>
> An intriguing finding from the above experiments is that the inference capabilities of OPT are **positively** affected by directly ensembling tokens on OPT on the relatively **larger** OPT6.7B model while **negatively** affected by the same setting on the relatively **smaller** OPT2.7B model. This observation provides a very useful guideline on where to apply CrossGET to accelerate LLM; that is, model size should be taken into consideration. A possible explanation for the contrasting behaviors is:
>
> - To accelerate OPT, the smaller 2.7B model is more vulnerable to the **disturbance brought by the token ensemble within the model** (note that the OPT model is **frozen**, so it cannot adapt its weights of parameters when the number of tokens is getting smaller). Therefore, reducing tokens on Q-Former is a better setting so that the OPT model is accelerated by taking fewer input tokens.
> - The larger 6.7B model is more resilient to the disturbance brought by the token ensemble within the model. Moreover, ensembling tokens within the OPT model can help **preserve the tokens’ information as much as possible** (if the number of tokens is reduced in the preceding Q-Former, the lost information due to the ensemble operation will be **inaccessible** to the succeeding OPT model). Moreover, the experiments also indicate that at least **two competing factors determine** the extent of the **image captioning performance affected**: 1) performance increases due to preserving more tokens’ information within the OPT model. 2) performance decreases depending on frozen OPT’s resilience to the disturbance brought by token ensemble within the model.
>
> ---
>
> Thanks again for Reviewer Uwik’s constructive suggestions and encouragement to help strengthen this work. We hope these explanations and experiments can resolve the concerns and help Reviewer 4d9L/Readers find this paper more positive. Please do not hesitate to let us know if there are new comments on this work or any further explanations we can provide.

---

### Official Review · Reviewer_4d9L · 2023-10-31

**Soundness:** 3 good
**Presentation:** 3 good
**Contribution:** 3 good
**Rating:** 6
**Confidence:** 5

**Summary:**

This paper introduces CrossGET, a token reduction-based strategy, to accelerate vision-language transformers. The key contributions of CrossGET can be summarized as follows: 1) CrossGET incorporates cross-modal guided information through cross-modal tokens. 2) CrossGET employs the Complete-Graph Soft Matching (CGSM) strategy, which offers more reliable token-matching results compared to existing bipartite soft matching strategies. Experimental evaluations conducted across multiple models, datasets, and tasks demonstrate the superior performance of the proposed method.

**Strengths:**

The acceleration of VL models is highly relevant for their practical deployment.

**Weaknesses:**

While this paper presents promising results and extensive evaluations, there are important concerns that should be addressed before publication.
1. Some experimental results are perplexing. Table 1 suggests that ToMe performs worse when equipped with Adapter or ExtraToken. However, Adapter and VPT are parameter-efficient tuning methods that enhance performance with minimal additional parameters. It is unclear how they could instead degrade performance. I suspect there may be errors in the implementations. It is recommended to double-check the results or provide convincing explanations. Additionally, the upper-right subfigure in Figure 4 is also confusing. In my understanding, CrossGET and ToMe have close GFLOPs under the same configuration (as evident from the left subfigure). Therefore, the significant differences in GFLOPs for each data point pair in the upper-right subfigure indicate that they are compared under different configurations. A reasonable explanation should be provided here. Moreover, the down-right subfigure seems to be unusual as well. How is it possible for the model to achieve even better performance (nearly 86) with only 1/10 GFLOPs? Are the settings the same as in other figures?

2. The contribution of the Complete-Graph Soft Matching (CGSM) appears to be minor. For instance, Table 1 suggests that ToMe and CrossGET $\Delta$ perform similarly in different metrics, indicating that the proposed CGSM may have little impact. ToMe employs the bipartite soft matching strategy for its efficiency and simplicity, and the ToMe paper demonstrates that this strategy can approximate optimal matching through extensive combination experiments. This paper should provide more evidence (visualizations, analytical experiments) to justify the effectiveness of the proposed CGSM.

3. Most experiments in this paper focus on Image-Text retrieval tasks. Is the proposed method equally effective in other VL tasks, such as the CoOP benchmark or open vocabulary segmentation?

4. This paper lacks an important comparison. [1] proposes reducing the number of tokens through clustering and demonstrates better performance than ToMe in accelerating transformers. However, this paper only briefly mentions it in the introduction without further discussion or comparisons. It is recommended to include more comparisons ([1] vs. CrossGET $\Delta$, [1] + CGM&CGE vs. CrossGET $\star$, etc., better in dense prediction tasks) with [1].

I am glad to increase my rating if my concerns are addressed.

[1]. Weicong Liang, Yuhui Yuan, Henghui Ding, Xiao Luo, Weihong Lin, Ding Jia, Zheng Zhang, Chao Zhang, and Han Hu. "Expediting large-scale vision transformer for dense prediction without fine-tuning." Advances in Neural Information Processing Systems, 35:35462–35477, 2022a.

**Questions:**

No other questions.

---

> ### Author Response · Authors · 2023-11-22
> **Response to Reviewer 4d9L [Part 1/3]**
>
> We sincerely appreciate Reviewer 4d9L’s thoughtful feedback. We are in accord with Reviewer 4d9L’s recognition that the proposed method has high practical value for VL models. Below we provide a detailed response with required experiments to address each concern.
>
> ---
>
> *W1. Some experimental results are perplexing.*
>
> A1.
>
> - For “*Table 1 suggests that ToMe performs worse when equipped with Adapter or ExtraToken*”: We have rechecked our implementations. It is worth noting that Table 1 actually suggests **ToMe performs better when equipped with the extra token**, which is as expected. However, Table 1 also suggests that ToMe performs worse when equipped with adapters because **ToMe uses full-parameter fine-tuning by default while only tuning a smaller number of parameters if adapters are used**. To elaborate, adapters freeze all the parameters except the newly added FFN branches. However, the token reduction methods will decrease the number of tokens and disturb modules whose number of tokens is reduced. Therefore, it is important to re-learn and update the weights of their attention modules. Experiments with adapters indicate that the lots of frozen parameters will damage the final performance of the token reduction method to some extent.
> - For *“The upper-right subfigure in Figure 4 is also confusing”*: Under the same configurations, CrossGET has slightly larger GFLOPs than ToMe since more complicated computations are conducted, and extra cross tokens are introduced. However, the slight discrepancy among the X-axes (GFLOPs) of different datapoints does not change the conclusion that CrossGET outperforms ToMe here, because **what matters most for model acceleration is the performance-cost trade-off**, *i.e.*, it is a binary relation always involved with performance and cost/speed simultaneously. Therefore, illustrating a **better curve** (the curve **lies above** the other) in trade-off plots is a widely recognized way to show the superiority of the proposed method. Besides, the X-axes of datapoints in the left subfigure are seemingly more consistent because their sampling interval is smaller, resulting in an illusion of dislocated alignment, *i.e.*, the datapoints which have the same X-axis values do not necessarily mean the same reduction ratios (but the ratios are indeed very close). Again, the conclusion that CrossGET outperforms ToMe remains unchanged, as evidenced by the most important relation that the curve of CrossGET lies above the curve of ToMe.
> - For *“The down-right subfigure seems to be unusual as well. How is it possible for the model to achieve even better performance (nearly 86) with only 1/10 GFLOPs”*: As explained by the caption of Figure 4, **these subfigures report performance on different models and tasks**. Moreover, different metrics are used as Y-axes. For example, the down-right subfigure illustrates the tradeoff for the CLIP model on the Flickr30K dataset of the Image-Text Retrieval task using average Top-1 recall as the metric. On the other hand, the left subfigure illustrates the tradeoff for the BLIP model on the NVLR2 dataset of the Visual Reasoning task using accuracy as the metric without training.
>
> ---

---

> ### Author Response · Authors · 2023-11-22
> **Response to Reviewer 4d9L [Part 2/3]**
>
> *W2. The contribution of the Complete-Graph Soft Matching (CGSM).*
>
> A2. We provide more evidence as follows to justify the effectiveness of the proposed CGSM.
>
> - Table1 in the paper demonstrates performance on the Image-Text Retrieval, which is a two-fold and multi-metric task. A more comprehensive and fair metric should be the average of the Image-to-Text Recall@1 (Top-1 Recall) and Text-to-Image Recall@1 (Top-1 Recall) as shown in Table 1:
>
>     | Approach | $\bar{R@1}$ | GFLOPs |
>     | --- | --- | --- |
>     | CLIP | 85.7 | 20.6 |
>     | ToMe | 84.5 | 11.8 |
>     | CrossGET(**CGSM**) | 85.0 (**+0.5**) | 11.9 |
>     | CrossGET(CGSM+**CGME**) | 85.9 (+0.5 **+0.9**) | 12.0 |
>
>     The above table indicates that on the CLIP model, the Complete-Graph Soft Matching (CGSM) gains a 0.5 improvement in the average of Top-1 recall metric, which accounted for about $0.5/1.4 \approx 36\\%$ of CrossGET's total improvement compared to ToMe (85.9-84.5=1.4).
>
> - Table2 in the paper demonstrates that Complete-Graph Soft Matching obtains consistent improvements across different tasks and models in addition to the previously described experiments.
>
>     | Approach | Test Acc | GFLOPs |
>     | --- | --- | --- |
>     | BLIP | 83.4 | 132.5 |
>     | ToMe | 82.2 | 59.0 |
>     | CrossGET(**CGSM**) | 82.6 (**+0.4**) | 60.8 |
>     | CrossGET(CGSM+**CGME**) | 83.2 (+0.4 **+0.6**) | 61.1 |
>
>     The above table shows that on the BLIP model, the Complete-Graph Soft Matching (CGSM) gains a 0.4 improvement in the Test Acc metric, which accounted for 0.4/1.0 = 40\\% of CrossGET's total improvement compared to ToMe (83.2-82.2=1.0).
>
> - The left subfigure of Figure 4 in the paper also presents that Complete-Graph Soft Matching obtains consistent improvements across different reduction ratios. Since the left subfigure reports performance without fine-tuning, the performance improvement over ToMe mainly gives the credit to Complete-Graph Soft Matching (CGSM).
>
> The above comparisons demonstrate that **the improvement given by Complete-Graph Soft Matching (CGSM) is not negligible** in the total improvement of CrossGET.
>
> ---
>
> *W3. Most experiments in this paper focus on Image-Text retrieval tasks.*
>
> A3.
>
> - We would like first to clarify that Image-Text retrieval is a small part of tasks that we have conducted experiments on. Apart from Image-Text Retrieval, we have also conducted experiments on the **Visual Reasoning** task in Table 2, the **Image Captioning** task in Tables 3, 6, and 23, the **Novel Object Caption** task in Table 4, and the **Visual Question Answer** task in Table 5.
> - We have added new experiments on CoOp benchmark as suggested by Reviewer 4d9L:
>     - Performance on the ImageNet dataset
>
>         | Method  | Top-1 Acc (%) | GFLOPs | Method  | Top-1 Acc (%) | GFLOPs |
>         | --- | --- | --- | --- | --- | --- |
>         | CoOp (16 shots) | 71.1 | 20.6 | CoOp (16 shots) | 71.1 | 20.6 |
>         | +ToMe | 70.4 | 16.2 | +CrossGET | **71.0**  | 16.5 |
>         | +ToMe | 68.9 | 14.0 | +CrossGET | **70.2** | 14.2 |
>         | +ToMe | 62.8 | 11.8 | +CrossGET | **67.4** | 12.0 |
>     - Performance on the Caltech101 dataset
>
>         | Method  | Top-1 Acc | GFLOPs | Method  | Top-1 Acc | GFLOPs |
>         | --- | --- | --- | --- | --- | --- |
>         | CoOp (16 shots) | 95.4 | 20.6 | CoOp (16 shots) | 95.4 | 20.6 |
>         | +ToMe | 93.7 | 14.0 | +CrossGET | **94.9** | 14.2 |
>         | +ToMe | 93.1 | 11.8 | +CrossGET | **94.0** | 12.0 |
>     - Performance on the OxfordPets dataset
>
>         | Method  | Top-1 Acc | GFLOPs | Method  | Top-1 Acc | GFLOPs |
>         | --- | --- | --- | --- | --- | --- |
>         | CoOp (16 shots) | 93.3 | 20.6 | CoOp (16 shots) | 93.3 | 20.6 |
>         | +ToMe | 89.5 | 14.0 | +CrossGET | **91.1** | 14.2 |
>         | +ToMe | 86.9 | 11.8 | +CrossGET | **89.6** | 12.0 |
>     - The entire CoOp benchmark contains 11 datasets while we conduct experiments on the first 3 datasets due to time constraints. We believe experiments on these 3 datasets are representative enough since the ImageNet dataset is the largest among 11 datasets, the Caltech101 dataset is middle-size, and the OxfordPets dataset is the smallest. We follow the same setting from CoOp that uses 16 shots and freezes the backbone model CLIP while conducting prompt tuning.
>     - The experimental results demonstrate that **CrossGET also consistently outperforms ToMe on the CoOp benchmark**. Besides, the performance-cost trade-off on the CoOp benchmark is relatively worse than other experiments we have reported in the paper, which should be attributed to:
>         - Most of the model parameters (*i.e.*, the whole backbone) are frozen, resulting in a worse convergence status than the full-parameter fine-tuning we have used for other experiments in the paper.
>         - Only a part of the datasets are used for few-shot learning, resulting in severer overfitting than the entire datasets we have used for other experiments in the paper.
>
> ---

---

> ### Author Response · Authors · 2023-11-22
> **Response to Reviewer 4d9L [Part 3/3]**
>
> *W4. An important comparison.*
>
> A4. We have added new comparison experiments with "[1] Expediting large-scale vision transformer for dense prediction without fine-tuning” as suggested by Reviewer 4d9L:
>
> | Method | Image to Text Top-1 Recall | Text to Image Top-1 Recall | Average Top-1 Recall | GFLOPs | Throughput |
> | --- | --- | --- | --- | --- | --- |
> | CLIP | 92.1 | 79.3 | 85.7 | 20.6 | 255.2 |
> | +Hourglass[1] | 90.5 | 77.9 | 84.2 | 15.0 | 342.3 |
> | +Hourglass[1] | 88.6 | 75.3 | 81.9 | 12.8 | 395.5 |
> | +CrossGET | **92.1** | **79.6** | **85.9** | 12.0 | 401.8 |
> - We conduct comparison experiments on the Image-Text Retrieval task and use CLIP as the baseline model. Even though Hourglass[1] shows impressive performance on segmentation tasks, the experimental results indicate that Hourglass[1] does not perform as well on the Image-Text Retrieval task as on segmentation tasks. We think this should be attributed to the fact that in dense prediction tasks, every token matters since the model is expected to output the corresponding category for each pixel. However, in the Image-Text Retrieval task, only very few tokens are crucial for the final output (*e.g.*, [CLS] token in the visual branch and [EOS] token in the language branch). Therefore, the reconstruction module used in Hourglass[1] becomes less important.
> - For “*better in dense prediction tasks*”: We are more than willing to investigate experiments on segmentation tasks. However, the obstacle is that the scope of this paper is to accelerate common general-purpose vision-language Transformers, e.g., CLIP, BLIP, and BLIP2-based models. However, these models do not implement segmentation tasks and provide pre-trained checkpoints for them. It is hard to reconstruct these models to fit in segmentation tasks and then train from scratch. We will be very interested in exploring multimodal segmentation acceleration if future general-purpose vision-language Transformers provide the necessary implementation and pre-trained checkpoints. But for now, we add experiments on the Image-Text Retrieval task instead to provide our initial exploration.
>
> ---
>
> Thanks again for Reviewer 4d9L’s constructive suggestions to help strengthen this work. We hope these explanations and experiments can resolve the concerns and help Reviewer 4d9L/Readers find this paper more positive. Please do not hesitate to let us know if there are new comments on this work or any further explanations we can provide.

---

### Official Review · Reviewer_oYGw · 2023-11-01

**Soundness:** 3 good
**Presentation:** 3 good
**Contribution:** 3 good
**Rating:** 6
**Confidence:** 4

**Summary:**

This paper proposes cross guided matching and cross guided ensemble as cross-modal importance indicator. Besides, a Complete-Graph Soft Matching algorithm is proposed as an improved version of ToME's bipartite soft matching.

**Strengths:**

1. Both Cross Guided Matching (CGM) and Complete-Graph Soft Matching (CGSM) is well motivated and proved to be effective.
2. Extensive experiments are conducted on several vision language tasks for both modal indenpendent VL model (CLIP) and modal dependent VL model (BLIP2). I do recognize the amount of work that went into this submission.

**Weaknesses:**

1. The proposed approach is named as Cross-Guided Ensemble of Tokens, however, I find that the proposed Cross-Guided Ensemble (CGE) is not that useful as illustrated in Table 1. So, I think the paper should re-organize the structure and highlight the really useful designs.
2. The proposed Complete-Graph Soft Matching is not specialized for cross-modal tasks, so does it outperform the ToMe algorithm in general visual recognition tasks?

**Questions:**

The proposed method can improve the model efficiency after training with little performance loss, and I am curious if the proposed method can also accelerate the training of multi-modal tasks.

---

> ### Author Response · Authors · 2023-11-22
> **Response to Reviewer oYGw [Part 1/2]**
>
> We sincerely appreciate Reviewer oYGw’s thoughtful feedback. We are glad to receive recognition from Reviewer oYGw that the proposed methods are well-motivated and have been widely validated. Below we provide a detailed response with required experiments to address each concern.
>
> ---
>
> *W1. The proposed approach is named as Cross-Guided Ensemble of Tokens, however, I find that the proposed Cross-Guided Ensemble (CGE) is not that useful as illustrated in Table 1.*
>
> A1.
>
> - We would like to clarify that the name “Cross-Guided Ensemble of Tokens” of the proposed approach does not represent that the Cross-Guided Ensemble (CGE) is the most crucial contributor to model performance. To elaborate, the **“Cross-Guided”** indicates we **utilize cross-modal guidance by Cross-Guided Matching and Ensemble(CGME)**, and the **“Ensemble”** emphasizes **the way to fulfill ensemble is important**, which is also the place that **Cross-Guided Soft Matching (CGSM) pitches in**. It is CGSM and CGME who work closely together that actually contribute to performance improvement.
> - For “*I find that the proposed Cross-Guided Ensemble (CGE) is not that useful as illustrated in Table 1”*: Table1 in the paper demonstrates performance on the Image-Text Retrieval, which is a multi-metric task. With a closer look at each metric, we can find that Cross-Guided Ensemble (CGE) contributes significantly to Top-5 and Top-10 Recall metrics:
>
>
>     | Method | Image to Text Top-5 Recall  | Image to Text Top-10 Recall  | Text to Image Top-5 Recall  | Text to Image Top-10 Recall  | GFLOPs |
>     | --- | --- | --- | --- | --- | --- |
>     | CLIP | 99.1 | 99.7 | 95.7 | 98.0 | 20.6 |
>     | CrossGET (CGM) | 99.3 | 99.7 | 95.3 | 97.7 | 12.0 |
>     | CrossGET (CGM + CGE) | 99.7 | 99.8 | 95.7 | 98.0 | 12.0 |
>
>     The above comparisons demonstrate that **the improvement given by Cross-Guided Ensemble (CGE) is not negligible** in the total improvement of CrossGET.
>
>
> ---
>
> *W2. Comparison with ToMe in general visual recognition tasks.*
>
> A2. We have added new experiments on the Image Classification task as suggested by Reviewer oYGw:
>
> - Comparison experiments on the ImageNet dataset
>
>
>     | Method  | Top-1 Acc (%) | GFLOPs | Method  | Top-1 Acc (%) | GFLOPs |
>     | --- | --- | --- | --- | --- | --- |
>     | CoOp  | 71.1 | 20.6 | CoOp (16 shots) | 71.1 | 20.6 |
>     | +ToMe | 70.4 | 16.2 | +CrossGET | **71.0**  | 16.5 |
>     | +ToMe | 68.9 | 14.0 | +CrossGET | **70.2** | 14.2 |
>     | +ToMe | 62.8 | 11.8 | +CrossGET | **67.4** | 12.0 |
> - Comparison experiments on the Caltech101 dataset
>
>
>     | Method  | Top-1 Acc | GFLOPs | Method  | Top-1 Acc | GFLOPs |
>     | --- | --- | --- | --- | --- | --- |
>     | CoOp  | 95.4 | 20.6 | CoOp (16 shots) | 95.4 | 20.6 |
>     | +ToMe | 93.7 | 14.0 | +CrossGET | **94.9** | 14.2 |
>     | +ToMe | 93.1 | 11.8 | +CrossGET | **94.0** | 12.0 |
> - Comparison experiments on the OxfordPets dataset
>
>
>     | Method  | Top-1 Acc | GFLOPs | Method  | Top-1 Acc | GFLOPs |
>     | --- | --- | --- | --- | --- | --- |
>     | CoOp  | 93.3 | 20.6 | CoOp (16 shots) | 93.3 | 20.6 |
>     | +ToMe | 89.5 | 14.0 | +CrossGET | **91.1** | 14.2 |
>     | +ToMe | 86.9 | 11.8 | +CrossGET | **89.6** | 12.0 |
> - We conduct experiments on 3 common datasets for the Image Classification task, including ImageNet, Caltech101, and OxfordPets. We follow the same few(16)-shot settings used in CoOp to train the model by prompt tuning. The experimental results demonstrate that **CrossGET also consistently outperforms ToMe in general visual recognition tasks**.
>
> ---

---

> ### Author Response · Authors · 2023-11-22
> **Response to Reviewer oYGw [Part 2/2]**
>
> *Q1. If the proposed method can also accelerate the training of multi-modal tasks.*
>
> A1. Yes, the proposed method **can also accelerate the training of multi-modal tasks**.
>
> | Method | $\bar{R@1}$ | GFLOPs | Inference Speedup | Training Speedup |
> | --- | --- | --- | --- | --- |
> | CLIP | 85.7 | 20.6 | 1x | 1x |
> | CLIP+CrossGET | 85.9 | 12.0 | 1.57x | 1.20x |
>
> | Method | Test Acc | GFLOPs | Inference Speedup | Training Speedup |
> | --- | --- | --- | --- | --- |
> | BLIP | 83.4 | 132.5 | 1x | 1x |
> | BLIP+CrossGET | 83.2 | 61.1 | 1.93x | 1.30x |
> - The above tables show examples of CLIP and BLIP models and demonstrate that acceleration during training is also achieved even though the speedup ratio is smaller than during inference. The reason is that the major acceleration originates from the forward stage, and **the speedup ratio gets smaller if the backward stage is involved** during training.
> - Furthermore, the above speedup ratio during training is reported with an unchanged batch size. In practical usage, less token number also means lower GPU memory consumption. As a result, we can use a larger batch size to achieve a higher speedup ratio during training than the above table reported.
>
> ---
>
> Thanks again for Reviewer oYGw’s constructive suggestions to help strengthen this work. We hope these explanations and experiments can resolve the concerns and help Reviewer oYGw/Readers find this paper more positive. Please do not hesitate to let us know if there are new comments on this work or any further explanations we can provide.

---

### Official Review · Reviewer_Yt1v · 2023-11-01

**Soundness:** 2 fair
**Presentation:** 3 good
**Contribution:** 2 fair
**Rating:** 5
**Confidence:** 4

**Summary:**

The paper proposes CrossGET to accelerate VLM by token merging. Specifically, this work introduces complete-graph matching to partition tokens and merge/reduce tokens based on similarities. The experimental results on common vision-language tasks demonstrate some effectiveness of the proposed method.

**Strengths:**

The paper is well-organized and the presentation is good. The motivation of accelerating VLMs is clear.

**Weaknesses:**

1. The major issue is novelty. CrossGET is incremental over ToMe by replacing ToMe's matching algorithm, adding learnable tokens and adapt unimodal ToMe to the multimodal setting.
2. As shown in Table 1, the newly proposed matching algorithm has marginal improvements.
3. CrossGET is proposed to accelerate heavy VLMs. However, majority of experiments are carried out on relatively light-weighted BLIP. There's only a small section for the truly heavy BLIP2, which is a stronger VLM that really needs acceleration.
4. CrossGET requires fine-tuning of VLMs. (1) In most cases, when models need fine-tuning, they are relatively small (acceleration is not demanding). (2) Huge VLMs that are really heavy can be used as zero-shot in different tasks or different datasets of a same task. In this sense, CrossGET which does not apply to pre-training stage is a bottleneck.
5. The paper fails to compare or adapt relevant works [1][2].

[1] DynamicViT: Efficient Vision Transformers with Dynamic Token Sparsification, NeurIPS 2021

[2] Not all patches are what you need: Expediting vision transformers via token reorganizations. ICLR 2022

**Final recommendation**: I agree the paper is improved by additional experiments and extensive analysis, and thus I raise my rating to 5.

**Questions:**

When CrossGET is applying to Flamingo or BLIP2 which uses frozen LLMs, it reduces to accelerating only vision encoders? Then, there will be a bunch of alternative approaches in accelerating ViTs?

---

> ### Author Response · Authors · 2023-11-22
> **Response to Reviewer Yt1v [Part 1/4]**
>
> We sincerely appreciate Reviewer Yt1v’s thoughtful feedback. We are more than glad to see Reviewer Yt1v recognizing this paper’s writing, presentation, and motivation. Below we provide a detailed response with required experiments to address each concern.
>
> ---
>
> *W*1. *About novelty.*
>
> A1. We would like to highlight the principal novelty of the proposed method and the crucial discrepancy with the existing method ToMe by answering the following two questions:
>
> - Why is the proposed Complete-Graph Soft Matching (CGSM) non-trivial?
>
>     The **superficial** advantage of the proposed CGSM over the existing Bipartite Soft Matching is that CGSM can produce more reliable token-matching results since each token takes into account the similarity with all other tokens but not half of the other tokens. However, A **vital challenge** behind the change from “Bipartite” to “Complete-Graph” lies in **how to ensure the parallelizability of the algorithm**. The feasible solution given by CGSM is **far beyond** simply extending the definition domains of the set of source tokens and the set of target tokens to all tokens. To elaborate, it is straightforward for Bipartite Soft Matching to ensure parallelizability since its set of source tokens and target tokens are naturally disjoint. In contrast, the definition domains of the set of source tokens and target tokens for CGSM are **coupled**. Therefore, the algorithm built on “Complete-Graph” is **originally non-parallelizable**. We must highlight that **it is not because of more reliable token-matching results but because of obtaining more reliable token-matching results WHILE maintaining parallelizability for high efficiency that reflects the proposed algorithm CGSM’s contribution and novelty**. On top of that, we also provide detailed discussions on the expectation of optimal matching probability and sub-optimal cases to substantiate the claims of more reliable token-matching results as well as the guarantee of parallelizability. Closely united empirical results and theoretical analysis make the proposed CGSM non-trivial.
>
> - Why is the proposed Cross-Guided Matching and Ensemble (CGME) non-trivial?
>
>     We would like to highlight that the proposed Cross-Guided Matching & Ensemble (CGME) is **the first method** to **make it possible** that **in modality-dependent models, the preceding modality can utilize information from the succeeding modality to guide token reduction**, which is a significant contribution considering most of the recent popular VLMs are exactly modality-dependent models (*e.g.*, InstructBLIP, MiniGPT4, and LLaVA). Moreover, we contribute **thorough empirical studies** to **provide detailed guidelines** on how to initialize, align, and utilize the cross tokens, the agents of cross-modal guidance, to further endow the underlying matching algorithm CGSM with the **capacity of perceiving and taking advantage of cross-modal information** in all kinds of model architectures.
>
> Moreover, we provide the following table to **distinguish the unique novelty and contributions of CrossGET** when compared with ToMe:
>
> | Novelty and Contributions  | ToMe | CrossGET | Remarks |
> | --- | --- | --- | --- |
> | 1. The underlying matching mechanism (empirical) | Bipartite Graph-based | Complete Graph-based: more reliable matching results while **maintaining parallelizability** | The achievement of the parallelizability in CrossGET is **far beyond** simply extending the definition domains of Bipartite Graph. |
> | 2. The underlying matching mechanism (theoretical) | N/A | 1. Expectation of optimal matching **probability**. 2. Discussion on **sub-optimal** cases. | Rigorous theoretical analyses **substantiate the claims** of more reliable token-matching results as well as the guarantee of parallelizability. |
> | 3. The high-level mechanism of utilizing cross-modal guidance (methodology) | N/A | Introducing cross tokens as agents to provide **cross-modal guidance** free from inference | It is the **first** method to **make it possible** that in **modality-dependent** models, the **preceding modality** can utilize information from the **succeeding modality** to guide token reduction. |
> | 4. The high-level mechanism of utilizing cross-modal guidance (empirical) | N/A | Thorough exploration of how to **initialize, align, and utilize** cross tokens | Providing **detailed guidelines** for practical usage in all kinds of model architectures. |
> | 5. Application (scope) | On uni-modal ViTs | On **multimodal** VLMs  | Including vision **LLMs**. |
> | 6. Experiment (performance) | As a strong baseline | Consistently **better** performance-cost **trade-offs** | Both **with** fine-tuning and **without** fine-tuning. |
>
> ---

---

> ### Author Response · Authors · 2023-11-22
> **Response to Reviewer Yt1v [Part 2/4]**
>
> *W2. Table 1 shows marginal improvements of the new matching algorithm.*
>
> *A2.* We provide more evidence as follows to justify the effectiveness of the proposed new matching algorithm CGSM.
>
> - Table1 in the paper demonstrates performance on the Image-Text Retrieval, which is a two-fold and multi-metric task. A more comprehensive and fair metric should be the average of the Image-to-Text Recall@1 (Top-1 Recall ) and Text-to-Image Recall@1 (Top-1 Recall) as shown in Table 1:
>
>
>     | Approach | $\bar{R@1}$ | GFLOPs |
>     | --- | --- | --- |
>     | CLIP | 85.7 | 20.6 |
>     | ToMe | 84.5 | 11.8 |
>     | CrossGET(**CGSM**) | 85.0 (**+0.5**) | 11.9 |
>     | CrossGET(CGSM+**CGME**) | 85.9 (+0.5 **+0.9**) | 12.0 |
>
>     The above table indicates that on the CLIP model, the Complete-Graph Soft Matching (CGSM) gains a 0.5 improvement in the average of Top-1 recall metric, which accounted for about $0.5/1.4 \approx 36\\%$ of CrossGET's total improvement compared to ToMe (85.9-84.5=1.4).
>
> - Table2 in the paper demonstrates that Complete-Graph Soft Matching obtains consistent improvements across different tasks and models in addition to the previously described experiments.
>
>
>     | Approach | Test Acc | GFLOPs |
>     | --- | --- | --- |
>     | BLIP | 83.4 | 132.5 |
>     | ToMe | 82.2 | 59.0 |
>     | CrossGET(**CGSM**) | 82.6 (**+0.4**) | 60.8 |
>     | CrossGET(CGSM+**CGME**) | 83.2 (+0.4 **+0.6**) | 61.1 |
>
>     The above table shows that on the BLIP model, the Complete-Graph Soft Matching (CGSM) gains a 0.4 improvement in the Test Acc metric, which accounted for 0.4/1.0 = 40% of CrossGET's total improvement compared to ToMe (83.2-82.2=1.0).
>
> - The left subfigure of Figure 4 in the paper also presents that Complete-Graph Soft Matching obtains consistent improvements across different reduction ratios. Since the left subfigure reports performance without fine-tuning, the performance improvement over ToMe mainly gives the credit to Complete-Graph Soft Matching (CGSM).
>
> The above comparisons demonstrate that **the improvement given by Complete-Graph Soft Matching (CGSM) is not negligible** in the total improvement of CrossGET.
>
> ---

---

> ### Author Response · Authors · 2023-11-22
> **Response to Reviewer Yt1v [Part 3/4]**
>
> *W3. There's only a small section for heavy BLIP2*
>
> A3.
>
> - We agree that heavy models need acceleration more eagerly than normal-size models. However, it does not mean the acceleration for normal-size models is unimportant. Accelerating normal-size models is also demanding for deploying models on consumer-level devices or low-performance cheap endpoints.
> - We are more than willing to continue our exploration of accelerating heavy VLMs, which is exactly at the top of our work list. For example, we have also added new experiments on heavy BLIP2 to investigate the impact of CrossGET on OPT in BLIP-2. We conduct new experiments on directly ensembling tokens on OPT. To elaborate, directly ensembling tokens on OPT leads to a **smaller KV cache**. Therefore, during each generation step, **a smaller number of previous tokens’ KV cache will attend to the current token** and thus need **less computational cost for Self-Attentions**, which is achieved by ensembling (fusing) previous tokens by Complete-Graph Soft Matching and cross-modal guidance from cross tokens.
>
>
>     | Model | Where to reduce tokens | CIDEr ($\uparrow$) | GFLOPs |
>     | --- | --- | --- | --- |
>     | BLIP2-OPT2.7B | / | 145.6 | 854.2 |
>     | BLIP2-OPT2.7B with CrossGET | On ViT and Q-Former | 143.1 | 413.9 |
>     | BLIP2-OPT2.7B with CrossGET | On ViT and LLM | 142.4 **(-0.7)** | 417.7 |
>     | BLIP2-OPT6.7B | / | 144.5 | 1042.6 |
>     | BLIP2-OPT6.7B with CrossGET | On ViT and Q-Former | 143.1 | 558.2 |
>     | BLIP2-OPT6.7B with CrossGET | On ViT and LLM | 143.5 **(+0.4)** | 566.1 |
>
>     An intriguing finding from the above experiments is that the inference capabilities of OPT are **positively** affected by directly ensembling tokens on OPT on the relatively **larger** OPT6.7B model while **negatively** affected by the same setting on the relatively **smaller** OPT2.7B model. This observation provides a very useful guideline on where to apply CrossGET to accelerate LLM; that is, model size should be taken into consideration. A possible explanation for the contrasting behaviors is:
>
>     - To accelerate OPT, the smaller 2.7B model is more vulnerable to the **disturbance brought by the token ensemble within the model** (note that the OPT model is **frozen**, so it cannot adapt its weights of parameters when the number of tokens is getting smaller). Therefore, reducing tokens on Q-Former is a better setting so that the OPT model is accelerated by taking fewer input tokens.
>     - The larger 6.7B model is more resilient to the disturbance brought by the token ensemble within the model. Moreover, ensembling tokens within the OPT model can help **preserve the tokens’ information as much as possible** (if the number of tokens is reduced in the preceding Q-Former, the lost information due to the ensemble operation will be **inaccessible** to the succeeding OPT model). Moreover, the experiments also indicate that at least **two competing factors determine** the extent of the **image captioning performance affected**: 1) performance increases due to preserving more tokens’ information within the OPT model. 2) performance decreases depending on frozen OPT’s resilience to the disturbance brought by token ensemble within the model.
>
> ---

---

> ### Author Response · Authors · 2023-11-22
> **Response to Reviewer Yt1v [Part 4/4]**
>
> *W4. Requiring fine-tuning of VLMs*
>
> A4. We would like to clarify that **fine-tuning is not a must for CrossGET**. In other words, **CrossGET can also be used without training** and still outperforms ToMe. First, the proposed Complete-Graph Soft Matching (CGSM) can be used without training. And the only part of CrossGET that requires training is learning the cross token. More specifically, we need to train the cross token to learn cross-modal information to better guide matching and ensemble of tokens. However, as we have discussed in Appendix C.5 of our paper, the cross token is initialized with informative features. Therefore, it can also be used to guide token matching and ensemble without training (Of course, untrained performance will be lower than trained, but it still demonstrates improvements over ToMe as shown in the left subfigure from Figure4 of our paper). **In cases where conducting zero-shot experiments or the cost of fine-tuning some huge VLMs is unaffordable, we can utilize CrossGET without fine-tuning, which still yields promising performance.**
>
> ---
>
> *W5. Compare or adapt DynamicViT and EViT.*
>
> A5. We have added new comparison experiments with DynamicViT and EViT as suggested by Reviewer Yt1v:
>
> | Method | Image to Text Top-1 Recall | Text to Image Top-1 Recall | Average Top-1 Recall | GFLOPs | Throughput |
> | --- | --- | --- | --- | --- | --- |
> | CLIP | 92.1 | 79.3 | 85.7 | 20.6 | 255.2 |
> | +DynamicViT  | 89.4 | 75.7 | 82.6 | 12.2 | 422.1 |
> | +EViT | 89.9 | 76.7 | 83.3 | 12.4 | 413.2 |
> | +CrossGET | **92.1** | **79.6** | **85.9** | 12.0 | 401.8 |
>
> The experimental results demonstrate the **superior performance of CrossGET compared with DynamicViT and EViT**.
>
> ---
>
> *Q1. When CrossGET is applying to Flamingo or BLIP2 which uses frozen LLMs, it reduces to accelerating only vision encoders? Then, there will be a bunch of alternative approaches in accelerating ViTs?*
>
> A1.
>
> - For “*When CrossGET is applying to Flamingo or BLIP2 which uses frozen LLMs, it reduces to accelerating only vision encoders?”*: **No, CrossGET accelerates vision encoders, Q-Former (if it exists), and LLMs simultaneously.** To elaborate, BLIP-2 consists of a ViT for processing visual input, a Q-Former(a Bert) for bridging modalities, and an LLM for taking inputs from Q-Former and generating text output accordingly. During fine-tuning, ViT and Q-Former are tunable while the LLM is frozen. To accelerate BLIP-2, we ensemble tokens in both ViT and Q-Former. As we explained in Appendix A.7, it is worth noting that there are **two ways to reduce the number of tokens processed by LLM**. The first one is reducing the number of tokens in Q-Former. Since LLM takes Q-Former’s output as its input, the **token reduction conducted in Q-Former leads to LLM acceleration**. The performance we reported in the paper is given by this setting. The second one is directly ensembling tokens in LLM. We have also tested this setting and discussed it in the answer to the previous question.
> - For *“Then, there will be a bunch of alternative approaches in accelerating ViTs?”*: There are alternative approaches to **accelerate uni-modal ViTs separately**, such as DynamicViT and EViT, that we have conducted experiments to compare. Moreover, experimental results demonstrate that **CrossGET consistently outperforms these alternative approaches in vision-language Transformers**.
>
> ---
>
> Thanks again for Reviewer Yt1v’s constructive suggestions to help strengthen this work. We hope these explanations and experiments can resolve the concerns and help Reviewer Yt1v/Readers find this paper more positive. Please do not hesitate to let us know if there are new comments on this work or any further explanations we can provide.

---

### Author Response · Authors · 2023-11-22
**General Response**

Dear Reviewers/AC/Readers interested in this paper:

We sincerely thank all reviewers for their invaluable time and constructive suggestions on our work, and we are deeply encouraged that reviewers recognize that the paper’s writing and presentation are good (Reviewer Yt1v), the motivation is clear (Reviewer Yt1v, oYGw), the proposed method is effective (Reviewer oYGw), thoroughly validated (Reviewer oYGw, Uwik), has solid theoretical foundation (Reviewer Uwik) and high practical value (Reviewer 4d9L, Uwik).

---

Here, we would like to highlight some vital novelty and contributions of the proposed approach:

- Why is the proposed Complete-Graph Soft Matching (CGSM) non-trivial?

    The **superficial** advantage of the proposed CGSM over the existing Bipartite Soft Matching is that CGSM can produce more reliable token-matching results since each token takes into account the similarity with all other tokens but not half of the other tokens. However, A **vital challenge** behind the change from “Bipartite” to “Complete-Graph” lies in **how to ensure the parallelizability of the algorithm**. The feasible solution given by CGSM is **far beyond** simply extending the definition domains of the set of source tokens and the set of target tokens to all tokens. To elaborate, it is straightforward for Bipartite Soft Matching to ensure parallelizability since its set of source tokens and target tokens are naturally disjoint. In contrast, the definition domains of the set of source tokens and target tokens for CGSM are **coupled**. Therefore, the algorithm built on “Complete-Graph” is **originally non-parallelizable**. We must highlight that **it is not because of more reliable token-matching results but because of obtaining more reliable token-matching results WHILE maintaining parallelizability for high efficiency that reflects the proposed algorithm CGSM’s contribution and novelty**. On top of that, we also provide detailed discussions on the expectation of optimal matching probability and sub-optimal cases to substantiate the claims of more reliable token-matching results as well as the guarantee of parallelizability. Closely united empirical results and theoretical analysis make the proposed CGSM non-trivial.

- Why is the proposed Cross-Guided Matching and Ensemble (CGME) non-trivial?

    We would like to highlight that the proposed Cross-Guided Matching & Ensemble (CGME) is **the first method** to **make it possible** that **in modality-dependent models, the preceding modality can utilize information from the succeeding modality to guide token reduction**, which is a significant contribution considering most of the recent popular VLMs are exactly modality-dependent models (*e.g.*, InstructBLIP, MiniGPT4, and LLaVA). Moreover, we contribute **thorough empirical studies** to **provide detailed guidelines** on how to initialize, align, and utilize the cross tokens, the agents of cross-modal guidance, to further endow the underlying matching algorithm CGSM with the **capacity of perceiving and taking advantage of cross-modal information** in all kinds of model architectures.

- Why the performance improvement over existing methods is non-trivial?

    We would like to explain why the performance improvements should be considered as **significant instead of marginal**. It is widely known that the model performance does **not improve linearly** with the growth of the computational cost. In other words, a slight improvement in accuracy is typically at the expense of notably huge computational costs. Take experiments on NLVR2 datasets in Table 2 of our paper as an example. CrossGET achieves nearly 2x speedup compared to the original model with only a 0.2% accuracy drop. Compared with the previous SOTA method ToMe, which has a 1.2% accuracy drop, CrossGET reduces the accuracy drop by more than $\frac{1.2-0.2}{1.2} \approx 83\\%$ while maintaining a similar acceleration speedup. We believe this improvement level should be considered as significant instead of marginal.


---

We have also addressed each reviewer's concerns in the individual posts and incorporated new results into the revised paper. We would greatly appreciate it if reviewers/AC could take a little time to check our responses.

Best wishes,

Paper6136 Authors